# Pupil dilation offers a time-window on prediction error

**Olympia Colizoli[1]\*, Tessa M van Leeuwen[1,2], Danaja Rutar[1,3], Harold Bekkering[1]**

[1]Donders Institute for Brain, Cognition and Behaviour, Radboud University, Nijmegen, Netherlands; [2]Department of Communication and Cognition, Tilburg University, Tilburg, Netherlands; [3]Department of Psychology, Sigmund Freud University, Ljubljana, Slovenia

## eLife Assessment

This **valuable** study investigates the relationship between pupil dilation and information gain during associative learning, using two different tasks. A key strength of this study is its exploration of pupil dilation beyond the immediate response period, extending analysis to later time windows after feedback, and it provides **convincing** evidence that pupillary response to information gain may be context-dependent during associative learning. The interpretation remains limited by task heterogeneity and unresolved contextual factors influencing pupil dynamics, but a range of interesting ideas are discussed.

**\*For correspondence:**
olympia.colizoli@donders.ru.nl

**Competing interest:** The authors declare that no competing interests exist.

**Abstract** Task-evoked pupil dilation is notably linked to unexpected events. Building on Zénon's (2019) information-theory framework, we investigated whether the pupil's response to feedback on decision outcomes during associative learning reflects a prediction error signal. Operationally, we defined prediction errors as an interaction between stimulus-pair frequency and accuracy. We then tested if these signals correlated with information gain, formally defined as the Kullback-Leibler (KL) divergence between posterior and prior belief distributions of an ideal observer. We reasoned that information gain should be proportional to the precision-weighted prediction error signals potentially arising from neuromodulatory arousal networks. We analyzed two data sets in which participants performed perceptual decision-making tasks while pupil dilation was recorded. Our findings consistently showed that a significant proportion of variability in the post-feedback pupil response was explained by information gain shortly after feedback presentation. For the first time, we present evidence that whether the pupil dilates or constricts along with information gain was context dependent. This study offers empirical evidence that the pupil's response provides valuable insights into the process of model updating during learning, highlighting its utility as a physiological indicator of internal belief states.

## Introduction

The human brain is constantly predicting its environment by forming expectations based on the meaningful environmental statistics we encounter in life (e.g. bananas are yellow, sometimes green, but never blue or red; *Aslin, 2017*; *Turk-Browne et al., 2005*; *Siegelman et al., 2019*). Furthermore, perception is largely driven by these expectations (*de Lange et al., 2018*; *Gau and Noppeney, 2016*). In the predictive processing framework, the brain constructs and updates internal models of the world that aim to accurately predict incoming sensory data by minimizing prediction errors (*Sprevak and Smith, 2023*). Prediction errors are abstractly defined as the difference between an expected and obtained outcome and are crucial concepts in models of learning (*den Ouden et al., 2012*; *Montague*

*et al., 2004*; *Schultz, 2006*). Pupil dilation may be a reliable biomarker of neural prediction error signals. Such a biomarker would be advantageous for investigating the neural computations involved in learning and decision-making due to the relative ease of measuring pupil size with a standard eye-tracking device. To achieve this aim, we must investigate the computational signatures reflected in pupil dilation to determine whether they genuinely represent prediction error signals during learning.

While reaction times (RT) scale with uncertainty, confidence, and reward expectation during perceptual decision-making (*Colizoli et al., 2018*; *de Gee et al., 2021*; *Ratcliff et al., 2016*; *Urai et al., 2017*), there is no overt behavioral marker that reflects the brain's processing of a prediction error following *feedback* on a decision outcome when the brain supposedly updates its internal model(s) of the world (*Sprevak and Smith, 2023*). Pupil dilation under constant luminance is a peripheral marker of the brain's central arousal system (*de Gee et al., 2017*; *Lloyd et al., 2023*; *McGinley et al., 2015*; *Murphy et al., 2014a*; *Reimer et al., 2016*). Evidence suggests that the brain leverages its arousal system to relay computational variables to circuits that execute inference and action selection (*Montague et al., 2004*; *de Gee et al., 2017*; *Aston-Jones and Cohen, 2005*; *Glimcher, 2011*; *Lak et al., 2017*; *Sara, 2009*; *Schultz, 2002*; *Yu and Dayan, 2005*). A growing body of literature suggests that pupil dilation may be a reliable physiological marker reflecting a prediction error signal in the post-feedback pupil response (*Colizoli et al., 2018*; *de Gee et al., 2021*; *Browning et al., 2015*; *Kayhan et al., 2019*; *Koenig et al., 2018*; *Nassar et al., 2012*; *O'Reilly et al., 2013*; *Preuschoff et al., 2011*; *Satterthwaite et al., 2007*; *Van Slooten et al., 2018*; *Rutar et al., 2023*). Whether a prediction error signal is detectable in pupil dilation will depend on the specific definition of 'prediction error', how it is operationalized, as well as the temporal nature of the signal itself. For instance, an unsigned prediction error is defined as the difference between expected and unexpected events but is agnostic to whether the events are better or worse than expected. In contrast, a signed prediction error indicates whether an outcome was better or worse than expected. Relatedly, reward prediction errors are a type of signed prediction error indicating whether an obtained reward was better or worse than expected (*den Ouden et al., 2012*).

In recent years, pupil dilation has received increased attention in psychology and human neuroscience research for its ability to reflect cognitive and computational variables involved in memory, attention, perception, and decision-making. For instance, pupil dilation has been shown to reflect stimulus expectancy and surprise (*de Gee et al., 2021*; *O'Reilly et al., 2013*; *Preuschoff et al., 2011*; *Alamia et al., 2019*; *Bianco et al., 2020*; *Friedman et al., 1973*; *Kamp and Donchin, 2015*; *Kloosterman et al., 2015*; *Knapen et al., 2016*; *Kuchinke et al., 2007*; *Lavín et al., 2013*; *Liao et al., 2016*; *Qiyuan et al., 1985*; *Raisig et al., 2010*; *Silvestrin et al., 2021*; *Wetzel et al., 2016*; *Zhao et al., 2019*; *Ghilardi et al., 2024*), decision uncertainty (*Colizoli et al., 2018*; *Urai et al., 2017*; *Nassar et al., 2012*; *O'Reilly et al., 2013*; *Van Slooten et al., 2018*; *Friedman et al., 1973*; *de Berker et al., 2016*; *Findling et al., 2019*; *Geng et al., 2015*; *Fan et al., 2023*; *Krishnamurthy et al., 2017*; *Lempert et al., 2015*; *Murphy et al., 2014b*; *Richer and Beatty, 1987*; *Vincent et al., 2019*), and the updating of belief states in internal models including (reward) prediction errors (*Colizoli et al., 2018*; *de Gee et al., 2021*; *Browning et al., 2015*; *Kayhan et al., 2019*; *Koenig et al., 2018*; *Nassar et al., 2012*; *O'Reilly et al., 2013*; *Preuschoff et al., 2011*; *Satterthwaite et al., 2007*; *Van Slooten et al., 2018*; *Lavín et al., 2013*; *Filipowicz et al., 2020*; *Harris et al., 2022*; *Pajkossy et al., 2023*; *Cheadle et al., 2014*; *He et al., 2024*). *Zénon, 2019* proposed that the plethora of cognitive phenomena reflected in pupil dilation can be unified under an information-theoretic framework. Under Zénon's hypothesis, the common factor driving all cognitive processes reflected in pupil dilation can be quantified in terms of information gain, defined as the divergence between posterior and prior beliefs. A unified framework that relates pupil dilation to cognition through information theory would be beneficial for several reasons. First, a unified framework would enable us to more accurately quantify cognitive processes by allowing us to connect physiological responses like arousal with cognitive functions. Second, such an approach could reveal how effectively an individual integrates new information and adjusts their predictions. Finally, a unified framework would allow researchers to apply consistent metrics across different contexts and tasks, facilitating comparisons between studies and enhancing our overall understanding of cognitive processes linked to pupil dilation.

We reasoned that the link between prediction error signals and information gain in pupil dilation is through precision weighting. Precision refers to the amount of uncertainty (inverse variance) of both the prior belief and sensory input in the prediction error signals (*Sprevak and Smith, 2023*; *Clark,*

*2017*; *Kwisthout et al., 2017*; *Iglesias et al., 2013*; *Yon and Frith, 2021*). More precise prediction errors receive more weighting and therefore have greater influence on model updating processes. The precision weighting of prediction error signals may provide a mechanism for distinguishing between known and unknown sources of uncertainty, related to the inherent stochastic nature of a signal versus insufficient information on the part of the observer, respectively (*Kwisthout et al., 2017*; *Yon and Frith, 2021*; *Press et al., 2020*). In Bayesian frameworks, information gain is fundamentally linked to prediction error, modulated by precision (*Kwisthout et al., 2017*; *Iglesias et al., 2013*; *Mathys et al., 2011*; *Yanagisawa et al., 2019*; *Kwisthout, 2017*; *Dijkstra et al., 2025*; *Smith et al., 2022*; *van Lieshout et al., 2025*; *Buckley et al., 2017*). In non-hierarchical Bayesian models, information gain can be derived as a function of prediction errors and the precision of the prior and likelihood distributions, a relationship that can be approximately linear (*Yanagisawa et al., 2019*). In hierarchical Bayesian inference, the update in beliefs (posterior mean changes) at each level is proportional to the precision-weighted prediction error; this update encodes the information gained from new observations (*Kwisthout et al., 2017*; *Iglesias et al., 2013*; *Mathys et al., 2011*; *Kwisthout, 2017*; *Dijkstra et al., 2025*). Neuromodulatory arousal systems are well-situated to act as precision weighting mechanisms in line with predictive processing frameworks (*Friston, 2008*; *Moran et al., 2013*). Empirical evidence suggests that neuromodulatory systems broadcast *precision-weighted* prediction errors to cortical regions (*de Gee et al., 2021*; *Harris et al., 2022*; *Iglesias et al., 2013*; *Haarsma et al., 2021*). Therefore, the hypothesis that feedback-locked pupil dilation reflects a prediction error signal is similarly in line with Zenon's main claim that pupil dilation generally reflects information gain, through precision weighting of the prediction error. We expected a prediction error signal in pupil dilation to be proportional to the information gain.

Information gain can be operationalized within information theory as the Kullback-Leibler (KL) divergence between the posterior and prior belief distributions of a Bayesian observer, representing a formalized quantity that is used to update internal models (*O'Reilly et al., 2013*; *Modirshanechi et al., 2023*; *Poli et al., 2024*). *Itti and Baldi, 2005* termed the KL divergence between posterior and prior belief distributions as 'Bayesian surprise' and showed a link to the allocation of attention. The KL divergence between posterior and prior belief distributions has been previously considered to be a proxy of (precision-weighted) prediction errors (*Press et al., 2020*; *Dijkstra et al., 2025*). According to Zénon's hypothesis, if pupil dilation reflects information gain during the observation of an outcome event, such as feedback on decision accuracy, then pupil size will be expected to increase in proportion to how much novel sensory evidence is used to update current beliefs (*O'Reilly et al., 2013*; *Zénon, 2019*). To our knowledge, there is a paucity of research on whether the pupil response is correlated with information gain, specifically focused on the interval following decision outcome/feedback presentation (*O'Reilly et al., 2013*; *Fleischmann et al., 2025*). Using a saccadic planning task, *O'Reilly et al., 2013* found that pupil dilation scaled negatively with information gain following target onset. While a significant correlation between post-target pupil dilation and information gain was obtained, the direction of this result seems at odds with the hypothesis that an increase in information gain would lead to greater pupil dilation, an issue we will return to in the Discussion. In contrast, *Fleischmann et al., 2025* reported a positive relationship between pupil dilation and information gain, which remained consistent across two auditory tasks requiring participants to predict either the temporal or spatial distributions of auditory sequences. Two other recent studies investigated a relationship between pupil dilation and information gain; however, the pupil dilation interval investigated occurred prior to information about decision outcome (*Zénon et al., 2024*; *Shirama et al., 2024*). *Zénon et al., 2024* similarly found that larger pupil responses were associated with more information gain with respect to the first operand during an arithmetic sum of two numbers. Finally, *Shirama et al., 2024* reported that the covariance between pupil and information gain depended on performance accuracy while participants predicted numbers in a changing environment. Specifically, in the low-performance group, pupil dilation positively tracked information gain. However, in the high-performance group, the direction was reversed. Taken together, there is evidence for both a positive and negative scaling between pupil dilation and information gain, depending on the task context and decision interval investigated.

The temporal dynamics of the relationship between the pupil response and information gain or prediction errors have not been consistently investigated. The temporal dynamics of prediction error signals in pupil dilation are likely informative because the brain's process of updating internal

models may contain an inherent temporal dimension (*Nienborg and Roelfsema, 2015*). Different temporal components of the pupil signal may correspond to different stages of predictive processing, for instance, as proposed by the hybrid predictive coding model (*Tscshantz et al., 2023*). These temporal dynamics can shed light on the mechanisms of predictive processing by clarifying the timing of learning updates in relation to feedback presentation (*Sales et al., 2019*). The temporal dynamics of prediction error signals may also help differentiate between types of learning, indicate how attention and cognitive load are allocated during tasks, and implicate specific brain regions or neural processes involved in learning (*Colantonio et al., 2023*; *Stemerding et al., 2022*). Previous studies have shown different temporal response dynamics of prediction error signals in pupil dilation following feedback on decision outcome: While some studies suggest that the prediction error signals arise around the peak (~1 s) of the canonical impulse response function of the pupil (*de Gee et al., 2021*; *Preuschoff et al., 2011*; *Lavín et al., 2013*; *Cheadle et al., 2014*; *He et al., 2024*; *Burlingham et al., 2022*), other studies have shown evidence that prediction error signals (also) arise considerably later with respect to feedback on choice outcome (*Colizoli et al., 2018*; *Browning et al., 2015*; *Van Slooten et al., 2018*; *Lavín et al., 2013*; *He et al., 2024*). A relatively slower prediction error signal following feedback presentation may suggest deeper cognitive processing, increased cognitive load from sustained attention or ongoing uncertainty, or that the brain is integrating multiple sources of information before updating its internal model. Taken together, the literature on prediction error signals in pupil dilation following feedback on decision outcome does not converge to produce a consistent temporal signature. The specific time window analyzed across different tasks can affect whether a prediction error signal is detected at all. The emergence of consistent results is important for validating the pupil as a biomarker of prediction error, by facilitating comparative research, informing predictive models, uncovering neural mechanisms, as well as improving practical applications. Many factors could potentially explain these discrepant results, such as different task contexts (e.g. stimulus modality, reward-based learning, probabilistic learning vs. perceptual discrimination), different approaches to the pupil analyses such as using simple contrasts or model-based regression, and different interpretations of what constitutes a 'prediction error'. Given these discrepancies, it is crucial to investigate the specific conditions under which pupil dilation reflects a (precision-weighted) prediction error.

## Aims of the current study

The current study was motivated by Zénon's hypothesis (*Zénon, 2019*) concerning the relationship between pupil dilation and information gain, particularly in light of the varying sources of signal and noise introduced by task context and pupil dynamics. By demonstrating how task context can influence which signals are reflected in pupil dilation, and highlighting the importance of considering their temporal dynamics, we aim to promote a more nuanced and model-driven approach to cognitive research using pupillometry. The literature summarized above prompted us to investigate whether the pupil's response to decision outcomes during learning aligns with a prediction error signal defined within an information-theoretic framework. While Zénon theoretically proposed a direct link between pupil dilation and information gain, this hypothesis has not been thoroughly tested in empirical studies. We sought to fill this gap in the literature and shed light on the relationship between information gain and uncertainty during learning as reflected in pupil dilation.

In the current study, we investigated whether the pupil's response to decision outcome (i.e. feedback) in the context of associative learning reflects a prediction error as defined operationally as an interaction between stimulus-pair frequency and accuracy, while also exploring the time course of this prediction error signal. Thereafter, we tested whether these prediction error signals correlated with information gain, defined formally as the KL divergence between posterior and prior belief distributions of the ideal observer. We reasoned that information gain should be proportional to the (precision-weighted) prediction error signals potentially arising from neuromodulatory arousal networks. To do so, we adapted a simple model of trial-by-trial learning of stimulus probabilities based on information theory from previous literature (*O'Reilly et al., 2013*; *Mars et al., 2008*; *Poli et al., 2020*). For completeness, Shannon surprise and entropy were also computed and related to the post-feedback pupil response. We analyzed two independent datasets featuring distinct associative learning paradigms, one characterized by increasing entropy and the other by decreasing entropy as the tasks progressed. By examining these different tasks, we aimed to identify commonalities (if any) in the results across varying contexts. Additionally, the contrasting directions of entropy in the two

tasks enabled us to disentangle the correlation between stimulus-pair frequency and information gain in the post-feedback pupil response.

In the first data set, participants were instructed to predict the upcoming orientation (left vs. right) of a visual target based on the probability of visual and auditory cues. In the second data set, participants were first exposed to letter-color pairs of stimuli in different frequency conditions during an odd-ball detection task. The letter-color pair contingencies were irrelevant to the odd-ball task performance. The participants subsequently completed a decision-making task in which they had to decide which letter was presented together most often with which color during the previous odd-ball detection task (match vs. no match). Pupil dilation was recorded during the decision-making tasks in both data sets, and the post-feedback pupil response was the event of interest. We did not formally compare the results across the two data sets given substantial differences between these two task contexts. We expected the post-feedback pupil dilation to scale with information gain in both tasks in a relatively early time window, following the results of O'Reilly et al. We explored whether later prediction error components in the post-feedback pupil dilation might reflect other information-theoretic variables, such as Shannon surprise or entropy.

To preview, the results show for the first time that whether the pupil dilates or constricts along with information gain was context dependent. Our findings are overall in line with Zénon's hypothesis that pupil dilation reflects information-theoretic processing and furthermore suggest that these signatures in pupil dilation are complex and multifaceted. This study provides empirical evidence that the pupil's response can shed light on model updating during learning, demonstrating the potential of this easily measured physiological indicator for exploring internal belief states.

## Results

The current study aimed to investigate whether the pupil's response to decision outcome in the context of associative learning reflects a prediction error (defined operationally as an interaction between stimulus-pair frequency and accuracy) and whether these prediction error signals correlate with information gain. In two independent data sets, the frequency of pairs of stimuli was modulated in different conditions to induce a gradient of uncertain states (associative learning) upon which the participants had to make decisions. Within each data set, we inspected task performance, the evoked pupil time course, and the averaged pupil response in an early as compared with a late time window. A signed prediction error was defined as the interaction between stimulus-pair frequency and performance accuracy averaged across conditions. Finally, we tested the linear relationship between the post-feedback pupil response and information gain in two ways: a trial-by-trial correlation analysis across the pupil time course and a complementary linear mixed model analysis within each time window of interest.

### Results from the cue-target 2AFC task (data set #1)

While pupil dilation was recorded, participants predicted the orientation (left or right) of the upcoming target stimulus (Gabor patch) based on the visual and/or auditory cues (*Figure 1A*). The target stimulus served as the feedback event of interest and was always presented after the participants made a prediction of the upcoming orientation of a Gabor patch by button press with the corresponding finger on the left or right hand. Dependent variables of interest were response accuracy, RT, and feedback-locked pupil response. Note that the results of the statistical tests are reported in the figure alongside the data illustrations (see *Figure 1*).

#### Behavioral performance

We evaluated the behavioral performance of the participants on the cue-target 2AFC task (data set #1). On average, participants were more accurate on the trials in the 80% condition as compared with the 20% condition tested with a Wilcoxon signed-rank t-test (*Figure 1B*). Notably, the average accuracy within these two frequency conditions approximated the frequency itself but with substantial individual differences across the sample (see individual lines in *Figure 1B*). An interaction between frequency condition and accuracy was obtained in RT tested with a two-way repeated-measures ANOVA (*Figure 1C*): post-hoc t-tests showed that participants were faster to respond on correct trials as compared with incorrect trials in the 80% condition, while in contrast, participants were slower

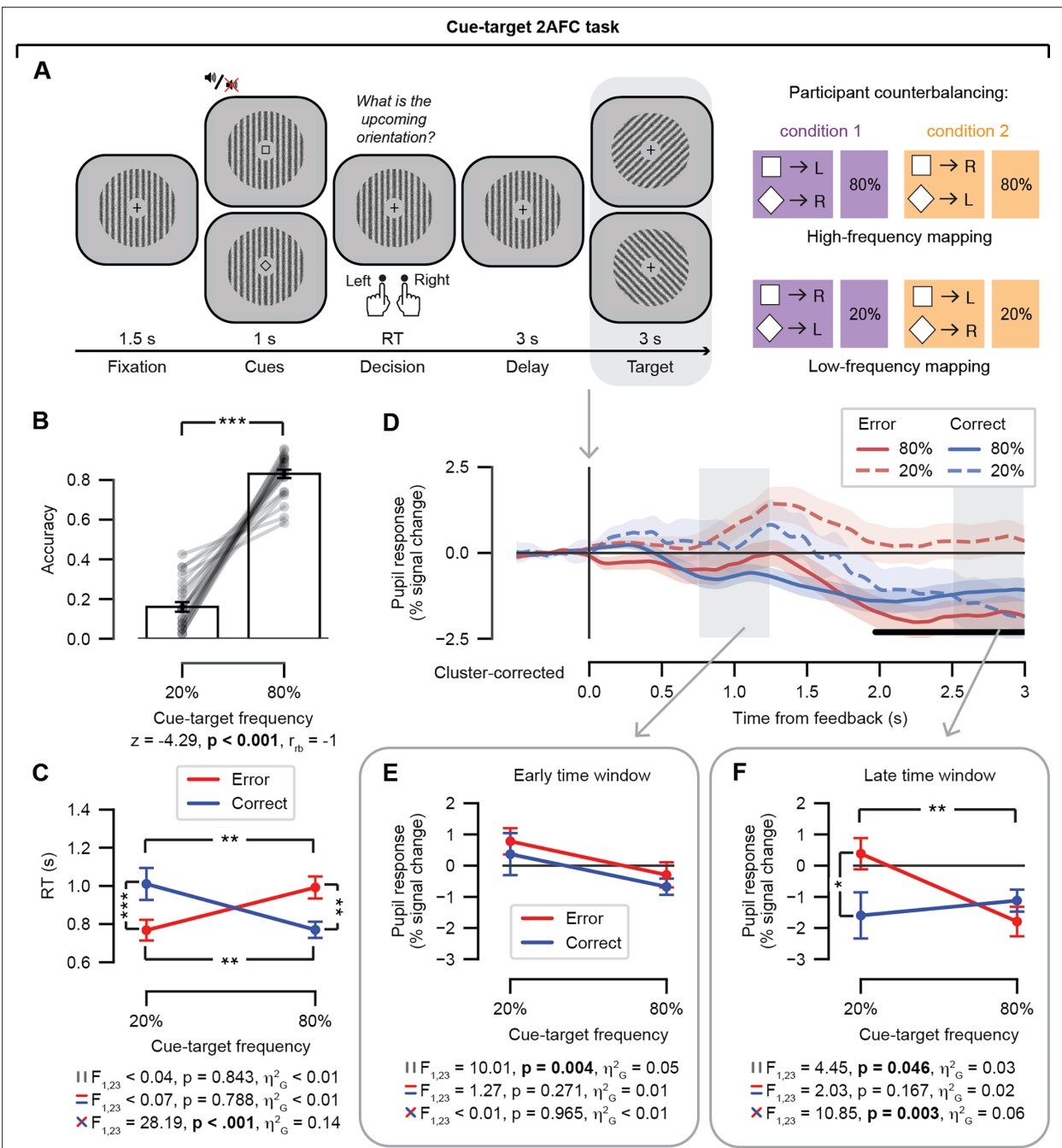

**Figure 1.** Data set #1: Cue-target 2AFC task and results. (**A**) Events during a single trial. While pupil dilation was recorded, participants predicted the orientation (left/right) of the upcoming target (Gabor patch) based on the visual and/or auditory cues (in the data analyzed here, only the visual cue had predictive validity). Predictions were given by a button press with the corresponding finger on the left or right hand. Two mapping conditions (condition 1 or condition 2) were counterbalanced across participants such that a participant in condition 1 was shown the square cue followed by a left-oriented target on 80% of the trials, while the square cue was followed by a right-oriented target on 20% of the trials. A gray box indicates the feedback event of interest. (**B**) Accuracy (fraction of correct responses) as a function of cue-target frequency. Data points are individual participants; stats, paired-samples Wilcoxon signed-rank t-test. (**C**) RT as a function of both cue-target frequency and accuracy (error/correct); stats, repeated-measures ANOVA. (**D**) Feedback-locked pupil response time course, plotted as a function of cue-target frequency and accuracy. Shading represents the standard error of the mean across participants (N = 24) . Light gray boxes, time windows of interest; early time window, [0.75, 1.25]; late time window, [2.5, 3.0]. The black horizontal bar indicates a significant interaction term (cluster-corrected, permutation test). (**E**) Early time window, average feedback-locked pupil response as a function of cue-target frequency and accuracy; stats, repeated-measures ANOVA. (**F**) As *E*, for the late time window. ANOVA results (multiple panels): top, main effect of frequency; middle, main effect of accuracy; bottom, frequency x accuracy interaction. Error bars, standard error of the mean across participants (N = 24). *p < 0.05, **p < 0.01, *** p < 0.001.

*Figure 1 continued on next page*

*Figure 1 continued*

The online version of this article includes the following source data and figure supplement(s) for figure 1:

**Source data 1.** Processed behavioral and pupil data used to generate the main results figure for the cue-target 2AFC task.

**Figure supplement 1.** Main effects of frequency and accuracy in the feedback-locked pupil time courses from the cue-target 2AFC task.

**Figure supplement 1—source data 1.** Processed pupil data used to generate the figure supplement showing main effects of frequency and accuracy in the feedback-locked pupil time courses from the cue-target 2AFC task.

to respond on correct trials as compared with incorrect trials for the 20% condition. We note that it was impossible for participants to determine from the cues whether the trial was in the 80% or 20% condition at the time when responses were given. Average RTs did not differ between frequency or accuracy conditions overall as indicated by a lack of main effects in the two-way ANOVA (*Figure 1C*).

## Comparing the feedback-locked pupil response between time windows

We were interested in evaluating the pupil's response to the feedback event: in the cue-target 2AFC task, feedback occurred with the presentation of the target stimulus following the prediction given by the participant with a button press. The pupil response time course locked to the onset of the target stimulus is shown in *Figure 1D* (gray boxes indicate the early and late time windows of interest) with the four conditions of interest defined by the two-way interaction between stimulus-pair frequency and accuracy plotted separately for the 3 s interval of target presentation. A significant interaction between frequency and accuracy emerged later in the trial around 2 s and was sustained until the next trial occurred (3 s; the black bar in *Figure 1D* refers to significant time points based on the cluster-based permutation test). The feedback-locked pupil response time courses are plotted for the main effects of stimulus-pair frequency and accuracy (see *Figure 1—figure supplement 1A–C*).

We formally tested for a difference on the average feedback-locked pupil response within the early as compared with late time windows in a three-way repeated-measures ANOVA with factors: time window (levels: early vs. late), frequency (levels: 20% vs. 80%), and accuracy (levels: error vs. correct). The results of the three-way repeated-measures ANOVA are presented in *Table 1*. Main effects of time window and frequency were obtained, in addition to a three-way interaction between the time window, frequency, and accuracy factors.

To break down the three-way interaction, the two-way interactions between stimulus-pair frequency and accuracy were tested in independent repeated-measures ANOVAs for the early and late time window (see *Figure 1E and F*, respectively). As suggested from the time course analysis (see *Figure 1D*), frequency and accuracy did not interact for the average feedback-locked pupil response within the early time window (*Figure 1E*). In contrast, the interaction between frequency and accuracy was significant for the average feedback-locked pupil response within the late time window

**Table 1.** Results of the three-way repeated-measures ANOVA on the feedback-locked pupil response in the cue-target 2AFC task (data set #1).

The three-way repeated-measures ANOVA included factors: time window (levels: early vs. late), frequency (levels: 20% vs. 80%), and accuracy (levels: error vs. correct).

| Factor | $F_{(1,23)}$ | p | $\eta^2_G$ |
|---|---|---|---|
| Time window | 7.52 | 0.012 | 0.05 |
| Frequency | 9.79 | 0.005 | 0.04 |
| Accuracy | 2.67 | 0.116 | 0.01 |
| Time window x frequency | 0.25 | 0.621 | <0.01 |
| Time window x accuracy | 0.26 | 0.614 | <0.01 |
| Frequency x accuracy | 3.45 | 0.076 | 0.02 |
| Time window x frequency x accuracy | 24.97 | <0.001 | 0.02 |

The online version of this article includes the following source data for table 1:

**Source data 1.** Processed behavioral data from the cue-target 2AFC task used for the three-way repeated-measures ANOVA included factors: time window (levels: early vs. late), frequency (levels: 20% vs. 80%), and accuracy (levels: error vs. correct).

(*Figure 1F*): post-hoc t-tests showed that the error trials drove the two-way interaction between frequency and accuracy such that pupils dilated more for error trials as compared with correct trials only in the 20% frequency condition; pupils also dilated more during errors in the 20% frequency condition than errors in the 80% condition. Taken together, the data suggest that the post-feedback pupil response may reflect unsigned prediction errors in the early time window and signed prediction errors in the late time window in the cue-target 2AFC task.

For both the early and late time windows, a main effect of frequency was obtained in each two-way ANOVA, while no main effect of accuracy was evident (see *Figure 1E and F*). Post-hoc t-tests showed that pupils dilated on average more for the 20% frequency condition (*M* = 0.57%, SE = 0.45) as compared with the 80% frequency condition (*M* = –0.49%, SE = 0.31) for the early time window. The frequency effect was in the same direction in the late time window, with larger pupil dilation for the 20% frequency condition (*M* = –0.61%, SE = 0.51) as compared with the 80% frequency condition (*M* = –1.46%, SE = 0.36). Larger pupil dilation in response to errors as compared with correct trials is a consistently reported effect in the literature (*Colizoli et al., 2018*; *de Gee et al., 2021*; *Urai et al., 2017*; *Braem et al., 2015*; *Critchley et al., 2005*; *Maier et al., 2019*; *Murphy et al., 2016*; *Rondeel et al., 2015*; *Wessel et al., 2011*); therefore, it is worth noting here that accuracy and frequency were highly correlated in the cue-target 2AFC task (see *Figure 1B*), which could explain the lack of a main effect of accuracy obtained here. This is further illustrated by comparing the 'Error' time course (see *Figure 1—figure supplement 1B*) with the '20%' time course (see *Figure 1—figure supplement 1C*). Likewise, compare the 'Correct' time course (see *Figure 1—figure supplement 1B*) with the '80%' time course (see *Figure 1—figure supplement 1C*). We note that within the early time window, the frequency effect in accuracy scaled with the frequency effect in the post-feedback pupil dilation across individual participants tested with a Spearman correction. Participants who showed a larger mean difference between the 80% as compared with the 20% frequency conditions in accuracy also showed smaller differences (a larger mean difference in magnitude in the negative direction) in pupil responses between frequency conditions (see *Appendix 1—figure 1*). This monotonic relationship between the frequency effect in accuracy and post-feedback pupil response indicates that the improvement in accuracy (as measured behaviorally) across trials was also reflected in the change in pupil dilation.

## Results from the letter-color 2AFC task (data set #2)

While pupil dilation was recorded, participants were administered a letter-color decision 2AFC task: participants indicated whether each letter presented 'matched' the subsequently presented colored square with a button press (*Figure 2A*; right-hand side). A match was correct when the letter and color had occurred most often together in the preceding odd-ball detection task. Dependent variables of interest were response accuracy, RT, and feedback-locked pupil response. Note that the results of the statistical tests are reported in the figure alongside the data illustrations (see *Figure 2*).

### Behavioral performance during the odd-ball detection task

The odd-ball detection task served as an independent learning phase for the letter-color pairs. Participants performed the task as expected: accuracy was lower in identifying trials with an odd-ball present (*M* = 84.0%, SD = 9.9) as compared to regular trials (*M* = 99.8%, SD = 0.3; $t_{46}$ = 11.04, p < 0.001, *d* = 1.61). Likewise, RTs were slower for trials with an odd-ball present (*M* = 0.49 s, SD = 0.01) as compared to regular trials (*M* = 0.42, SD = 0.01; $t_{46}$ = 16.53, p < 0.001, *d* = 2.41).

### Behavioral performance on the letter-color 2AFC task

We evaluated the behavioral performance of the participants on the letter-color 2AFC task (data set #2). During the letter-color 2AFC task, participants could accurately indicate whether a letter was presented most often with a given color in the preceding odd-ball detection task: response accuracy (around 80%) was higher than chance level (50%) in each of the three stimulus-pair frequency conditions on average (see bars in *Figure 2B*). This was also true for most participants (see individual lines in *Figure 2B*); however, neither accuracy (*Figure 2B*) nor RTs (*Figure 2C*) differed across the frequency conditions tested with repeated-measures ANOVAs (levels: 33%, 50%, and 84%). A main effect of accuracy as well as an interaction between frequency condition and accuracy was obtained in RT tested in a two-way repeated-measures ANOVA (*Figure 2C*). Post-hoc t-tests showed that participants were slower on error trials (*M* = 0.97 s, SE = 0.04) as compared with correct trials (*M* = 0.72 s,

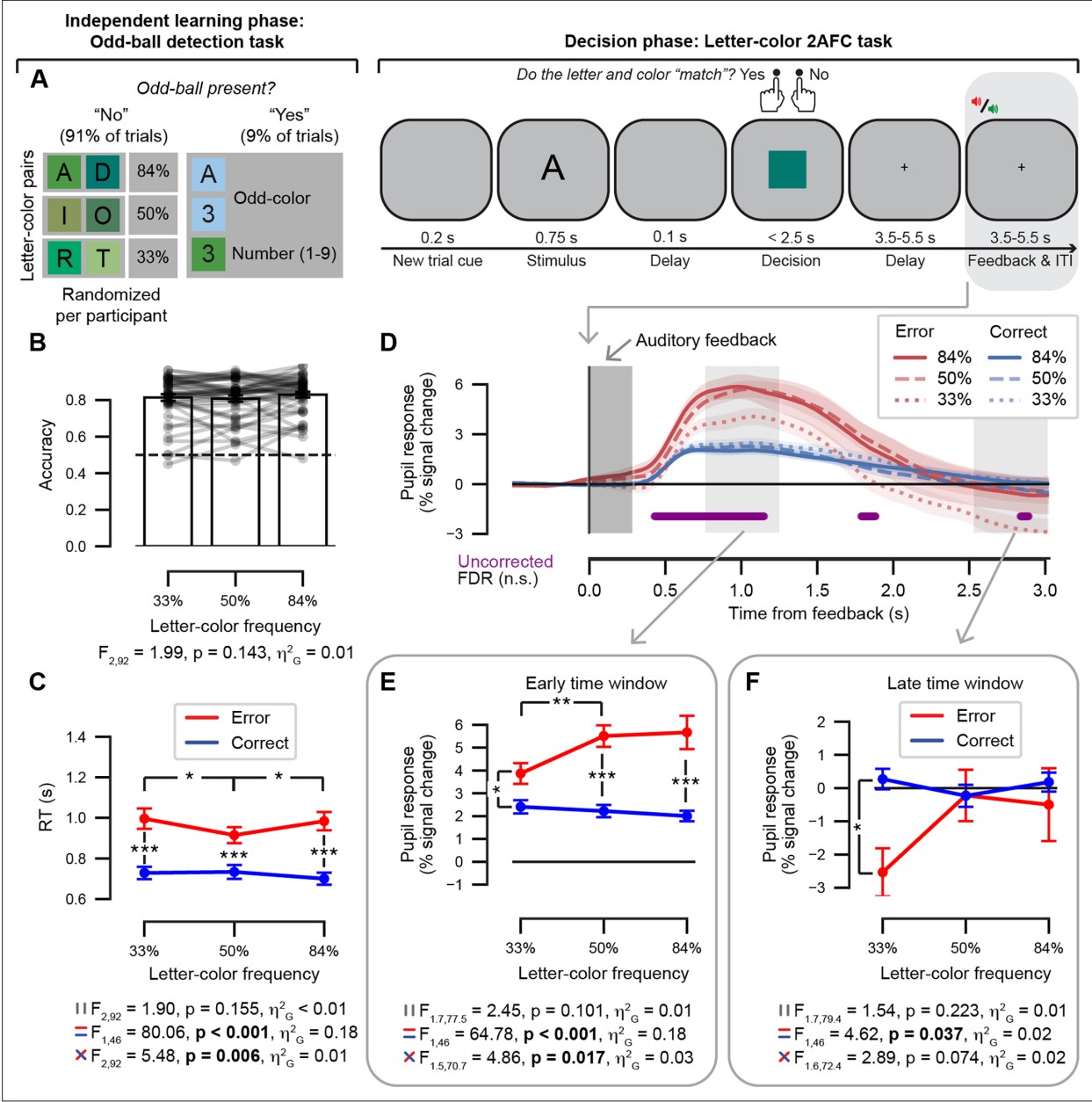

**Figure 2.** Data set #2: Letter-color 2AFC task and results. (**A**) *Left*, an independent learning phase was administered in the form of an odd-ball detection task during which six letters together with six shades of green colors as background (squares) were presented in three frequency conditions (33%, 50%, and 84%) on most trials (91%). Participants had to quickly respond to odd-ball targets (numbers and/or non-green color, 9% of trials). Letter-color mapping conditions were randomized per participant. *Right*, events during a single trial of the subsequent letter-color decision 2AFC task. While pupil dilation was recorded, participants indicated whether the letter "matched" the colored square with a button press. A match was correct when the letter and color had occurred most often together in the preceding odd-ball task. A gray box indicates the feedback event of interest. (**B**) Accuracy (fraction of correct responses) as a function of letter-color frequency. Dashed line represents chance level. Data points are individual participants; stats, repeated-measures ANOVA. (**C**) RT as a function of both letter-color frequency and accuracy; stats, repeated-measures ANOVA. (**D**) Feedback-locked pupil response time course, plotted as a function of letter-color frequency and accuracy. Shading represents the across participants of the mean (N = 47). Dark gray box, duration of the auditory feedback stimulus (0.3 s). Light gray boxes, time windows of interest; early time window, [0.75, 1.25]; late time window, [2.5, 3.0]. The purple horizontal bar indicates a significant two-way interaction effect (uncorrected for multiple comparisons). No significant time points remained after correction using the false discovery rate (FDR). (**E**) Early time window, average feedback-locked pupil response as a function of letter-color frequency and accuracy; stats, repeated-measures ANOVA (**F**) As *E*, for the late time window. ANOVA results (multiple panels): top, main effect of frequency; middle, main effect of accuracy; bottom, frequency x accuracy interaction. Error bars, standard error of the mean across participants (N = 47). *p < 0.05, **p < 0.01, *** p < 0.001.

The online version of this article includes the following source data and figure supplement(s) for figure 2:

*Figure 2 continued on next page*

*Figure 2 continued*

**Source data 1.** Processed behavioral and pupil data used to generate the main results figure for the letter-color 2AFC task.

**Figure supplement 1.** Main effects of frequency and accuracy in the feedback-locked pupil time courses from the letter-color 2AFC task.

**Figure supplement 1—source data 1.** Processed pupil data used to generate the figure supplement showing main effects of frequency and accuracy in the feedback-locked pupil time courses from the letter-color 2AFC task.

SE = 0.03). Breaking down the interaction in RT, post-hoc t-tests indicated that the accuracy difference between correct and error trials in the 50% frequency condition ($M$ = 0.18 s, SE = 0.03) was significantly smaller as compared with that for the 33% frequency condition ($M$ = 0.27 s, SE = 0.04; $z$ = –2.15, p = 0.031, $r_{rb}$ = 0.36) as well as the 84% frequency condition ($M$ = 0.28 s, SE = 0.04; $z$ = –2.55, p = 0.010, $r_{rb}$ = –0.43), while the accuracy difference between the 33% and 84% frequency conditions did not differ on average ($z$ = –0.78, p = 0.440, $r_{rb}$ = –0.13).

## Comparing the feedback-locked pupil response between time windows

We were interested in evaluating the pupil's response to the feedback event: in the letter-color 2AFC task, explicit feedback was administered on each trial in the form of an auditory tone (error vs. correct) following the prediction given by the participant with a button press. The pupil response time course locked to the onset of the auditory feedback stimulus is shown in *Figure 2D* (gray boxes indicated the early and late time windows of interest) with the six conditions of interest defined by the two-way interaction between stimulus-pair frequency and accuracy plotted separately for a 3 s post-feedback interval. A two-way repeated-measures ANOVA was computed independently for each time point in the feedback-locked pupil time course in *Figure 2D*. Clusters indicating an interaction between frequency and accuracy emerged at three distinct time points, but none of these clusters survived corrections for multiple comparisons using the False Discovery Rate (see purple bar, *Figure 2D*). The feedback-locked pupil response time courses are plotted for the main effects of stimulus-pair frequency and accuracy (see *Figure 2—figure supplement 1*). Using cluster-based permutation tests, we found that a robust main effect of accuracy spanned the early time window (see *Figure 2—figure supplement 1B*), while no main effect of frequency was obtained at any time points (see *Figure 2—figure supplement 1C*).

We formally tested for a difference in the average feedback locked pupil response averaged within the early as compared with late time windows in a three-way repeated-measures ANOVA with factors: time window (levels: early vs. late), frequency (levels: 33%, 50%, and 84%) and accuracy (levels: error vs. correct). The results of the three-way repeated-measures ANOVA are presented in *Table 2*. Main effects of time window and accuracy were obtained. The two-way interactions between time window

**Table 2.** Results of the three-way repeated-measures ANOVA on the feedback-locked pupil response in the letter-color 2AFC task (data set #2).

The three-way repeated-measures ANOVA included factors: time window (levels: early vs. late), frequency (levels: 33%, 50%, and 84%), and accuracy (levels: error vs. correct). Greenhouse-Geisser statistics are reported when assumptions of sphericity were violated.

| Factor | df | F | p | $\eta^2_G$ |
|---|---|---|---|---|
| Time window | (1, 46) | 130.16 | **<0.001** | 0.23 |
| Frequency | (1.639, 75.415) | 2.11 | 0.137 | 0.01 |
| Accuracy | (1, 46) | 4.76 | **0.034** | 0.01 |
| Time window x frequency | (2, 92) | 0.22 | 0.806 | <0.01 |
| Time window x accuracy | (1, 46) | 61.11 | **<0.001** | 0.06 |
| Frequency x accuracy | (1.480, 68.095) | 3.91 | **0.036** | 0.02 |
| Time window x frequency x accuracy | (2, 92) | 1.05 | 0.354 | <0.01 |

The online version of this article includes the following source data for table 2:

**Source data 1.** Processed behavioral data from the letter-color 2AFC task used for the three-way repeated-measures ANOVA included factors: time window (levels: early vs. late), frequency (levels: 20% vs. 80%), and accuracy (levels: error vs. correct).

and accuracy as well as frequency and accuracy were obtained. The three-way interaction between the time window, frequency, and accuracy factors was not significant.

We continued with a post-hoc exploration of the two-way interactions between stimulus-pair frequency (levels: 33%, 50%, 84%) and accuracy (levels: error vs. correct) by testing separate repeated-measures ANOVAs for the early and late time window given our a priori hypotheses about the nature of the three-way interaction. In the early time window, the two-way interaction between frequency and accuracy was obtained for the average feedback-locked pupil response (*Figure 2E*). To break down this two-way interaction between frequency and accuracy in the early time window (*Figure 2E*), we compared the difference between error as compared with correct trials across each pair of frequency conditions using t-tests: the accuracy difference in the 33% frequency condition (*M* = 1.46%, SE = 0.45) was significantly smaller as compared with that for the 50% frequency condition (*M* = 3.38%, SE = 0.43; $z = –2.94$, $p = 0.003$, $r_{rb} = –0.49$) as well as the 84% frequency condition (*M* = 3.66%, SE = 0.74; $z = –2.32$, $p = 0.020$, $r_{rb} = –0.39$), while the accuracy difference between the 50% and 84% frequency conditions did not differ on average ($z = 0.65$, $p = 0.525$, $r_{rb} = 0.11$). In the late time window, only a trend towards an interaction between frequency and accuracy was evident (*Figure 2F*). In both the early and late time windows, the two-way ANOVAs showed that a main effect of accuracy was obtained for the average feedback-locked pupil responses while no main effect of frequency was obtained. Interestingly, the direction of this main effect of accuracy differed per time window indicated by post-hoc t-tests (compare *Figure 2F* with *Figure 2G*): In the early time window, pupil dilation was larger on average for error trials (*M* = 5.01%, SE = 0.39) as compared with correct trials (*M* = 2.21%, SE = 0.23), while this effect reversed in direction during the late time window with larger pupil dilation for correct trials (*M* = 0.07%, SE = 0.26) as compared with error trials (*M* = –1.09%, SE = 0.55). Taken together, the data suggest that the post-feedback pupil response may reflect signed prediction errors, albeit more strongly within the early as compared with late time window; most striking is the fact that the direction of this interaction reversed across these time intervals.

In the letter-color 2AFC task, no scaling was evident between the frequency effect in accuracy and the frequency effect in the post-feedback pupil dilation across individual participants tested with a Spearman correlation (see *Appendix 1—figure 1*). In contrast to the cue-target 2AFC task, no relationship between behaviorally accuracy measures and changes in pupil dilation was obtained in the letter-color 2AFC task.

## Ideal learner model fits to feedback-locked pupil response

The ideal learner model used the stimulus information on each trial to estimate the information gain, surprise, and entropy during each of the decision-making tasks (see Materials and methods for further details). We fit the feedback-locked pupil response to the resulting model parameters in two complementary analyses: First, the relationship between the ideal learner models and the post-feedback pupil response was assessed by a correlation analysis (see *Figure 3* showing the average correlation coefficients across participants). Each time point in the pupil time course was correlated to each of the theoretic variables independently to see which signal the feedback-locked pupil response may be reflecting (if any) and furthermore to see how these patterns developed over time (within the 0–3 s window following feedback). Second, two linear (mixed) models were compared to see which combination of predictor variables best explained the feedback-locked pupil response while accounting for shared variance between predictor variables (see *Appendix 2—tables 1 and 2*).

### Ideal learner model fits for the cue-target 2AFC task: correlation analysis

The information-theoretic variables are shown as a function of task trial in *Figure 3A* and as a function of stimulus-pair frequency in *Figure 3B*. Multicollinearity was assessed by the correlations between the model parameters (information gain vs. surprise: $r = 0.09$, $p < 0.001$; information gain vs. entropy: $r = 0.36$, $p < 0.001$; surprise vs. entropy: $r = 0.17$, $p < 0.001$). We independently evaluated the trial-by-trial correlations between the information-theoretic variables and the post-feedback pupil response during the cue-target 2AFC task to investigate the variance explained by each of the three model parameters. The time course of the resulting correlation coefficients showed a pattern in which the pupil scaled negatively with information gain almost immediately after feedback onset until about 1 s into the feedback interval extending into the early time window (*Figure 3C*, purple line). The post-feedback pupil response also scaled with surprise starting around 0.5 s with respect to feedback

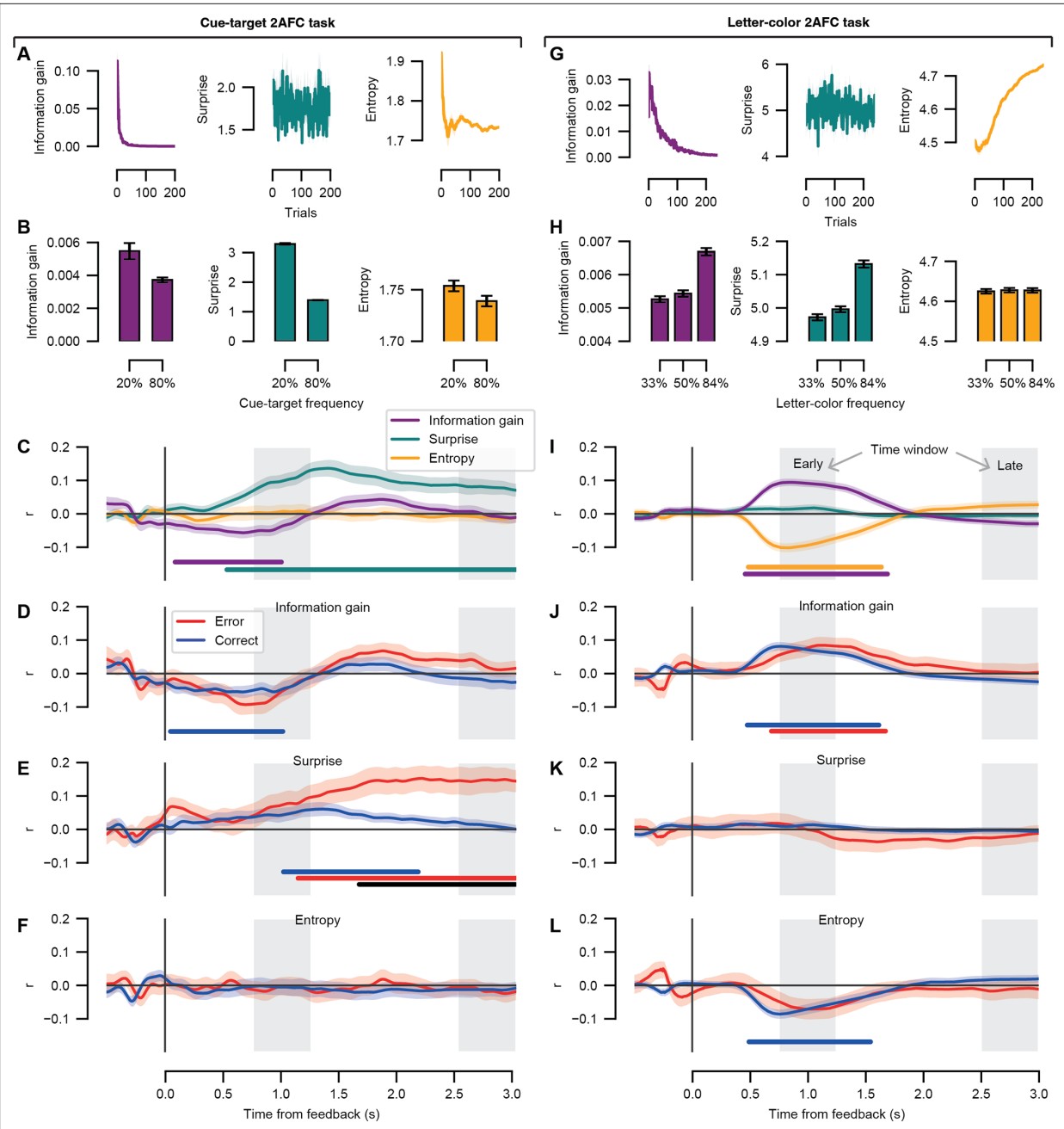

**Figure 3.** Correlations between the feedback-locked pupil response time course and the information-theoretic variables. *Left column*, results for the cue-target 2AFC task. *Right column*, results for the letter-color 2AFC task. (**A**) The information gain, surprise, and entropy parameters are shown as a function of task trial. Model parameter units are in bits. (**B**) The mean information gain, surprise, and entropy parameters are shown as a function of frequency condition. (**C**) Average trial-by-trial correlations at the group level between the ideal learner model parameters (information gain, surprise, and entropy) at each time point in the feedback-locked pupil response. (**D**) Average trial-by-trial correlations at the group level between the information gain parameter and the feedback-locked pupil response separately for the error and correct trials. (**E**) As *D*, for the surprise parameter. (**F**) As *D*, for the entropy parameter. (**G–L**) As *A-F* for the letter-color 2AFC task. (**C–L**) Shading represents the standard error of the mean across participants (cue-target 2AFC task: N = 24; letter-color 2AFC task: N = 47). Light gray boxes, time windows of interest; early time window, [0.75, 1.25]; late time window, [2.5, 3.0]. The colored horizontal bars indicate time periods of significant correlation coefficients tested against zero for each model parameter or condition of interest (cluster-corrected, permutation test). The black horizontal bar indicates a difference between conditions (cluster-corrected, permutation test).

The online version of this article includes the following source data and figure supplement(s) for figure 3:

**Source data 1.** Ideal learner model parameters and correlations between the feedback-locked pupil response time course and the information-theoretic variables (both tasks).

*Figure 3 continued on next page*

*Figure 3 continued*

**Figure supplement 1.** Correlations between the feedback-locked pupil response time course and the information-theoretic variables using a uniform prior distribution in the letter-color 2AFC task.

**Figure supplement 1—source data 1.** Ideal learner model parameters and correlations between the feedback-locked pupil response time course and the information-theoretic variables using a uniform prior distribution in the letter-color 2AFC task.

onset and lasted throughout the duration of the feedback interval, notably spanning both the early and late time windows (*Figure 3C*, teal line). No scaling between the post-feedback pupil response and entropy was obtained.

Note that this trial-by-trial analysis considers both error and correct trials simultaneously and is therefore not sensitive to the actual behavioral performance of each participant. To see whether the correlation between the information-theoretic variables and pupil response differed as a function of behavioral performance, we repeated the same correlation analysis of the model parameters to the feedback-locked pupil response separately for error and correct trials. Results showed that the post-feedback pupil response during correct trials scaled negatively with information gain from feedback onset until 1 s within the feedback interval (*Figure 3D*, blue line). The correlation between surprise and the feedback-locked pupil response for the error and correct trials diverged around 1.75 s with respect to feedback onset, and this difference persisted into the late time window (*Figure 3E*, black line). This suggests that Shannon surprise might be linked to the signed prediction errors (defined by the interaction between accuracy and stimulus-pair frequency) evident in the late time window of the cue-target 2AFC task. No scaling between the post-feedback pupil response and entropy was obtained for either correct or error trials (*Figure 3F*).

## Ideal learner model fits for the cue-target 2AFC task: linear mixed model analysis

The correlation analysis described above did not account for shared variance between model parameters. Therefore, to assess the unique variance explained by each of the model parameters during the cue-target 2AFC task, we compared two linear mixed models with the feedback-locked pupil response as the dependent variable (see Materials and methods section for further details). The model comparison was performed independently for the early and late time windows. For the cue-target 2AFC task, Model 1 performed significantly better than Model 2 for both the early ($R^2 = 0.29$, $\sigma = 4.28$) and late time window ($R^2 = 0.38$, $\sigma = 5.74$; see *Appendix 2—table 1* for the model comparison results), indicating that the inclusion of an interaction term between information gain and entropy did not lead to a better model fit of the post-feedback pupil response. The parameters of Model 1 are presented in *Table 3*. The linear mixed model analysis corroborated the correlation analysis: in the early time window, the predictor variables of surprise and information gain explained significant variance in the feedback-locked pupil response, while in the late time window, only the surprise predictor was significant. As expected, the pre-feedback baseline pupil dilation explained a significant amount of variance in the feedback-locked pupil response in both time windows. RT, however, did not explain significant variance in either time window.

When examining the correct and error trials separately, Model 2 never outperformed Model 1 (see *Appendix 2—table 1*); therefore, we only report the parameter estimates for Model 1 (*Table 3*). The linear mixed modeling results for the correct trials corroborated the correlation analysis: the early time window ($R^2 = 0.38$, $\sigma = 5.74$) showed that both the predictor variables of surprise and information gain explained significant variance in the feedback-locked pupil response; for correct trials in the late time window, ($R^2 = 0.33$, $\sigma = 6.00$), none of the information-theoretic predictor variables were significant. The pattern of linear mixed model results for the error trials differed somewhat from the correlation analysis in the early ($R^2 = 0.34$, $\sigma = 4.07$) and late time windows ($R^2 = 0.50$, $\sigma = 5.00$; compare *Table 3* with *Figure 3D and E*). First, in the early time window, a trend towards a scaling between information gain and the post-feedback pupil dilation across the early time window is apparent in the error trials in the correlation analysis (see *Figure 3D*). This trend became significant in the linear mixed modeling result: the predictor variable of information gain explained significant variance in the early feedback-locked pupil response for error trials. Second, in the late time window, error trials scaled with surprise in the correlation analysis (see *Figure 3E*), not entropy (see *Figure 3F*). In contrast, entropy explained

**Table 3.** Linear mixed model results for the cue-target 2AFC task.

Explanation of abbreviations, rows: *I*, Shannon surprise predictor variable; *H*, entropy; $D_{KL}$, information gain; *Baseline*, pre-feedback baseline pupil dilation; *RT*, reaction times. Columns: *95% CI*, the 95% credible interval of the median posterior distribution; *pd*, the probability (in percentage) of direction; *ESS*, effective sample size; *indicates strong evidence that the parameter has a positive/negative effect on the post-feedback pupil response.

| Cue-target 2AFC task | | | | | | | | | | |
|---|---|---|---|---|---|---|---|---|---|---|
| Early time window | | | | | | Late time window | | | | |
| | Parameter | Median | 95% CI | pd (%) | ESS | | Parameter | Median | 95% CI | pd (%) | ESS |
| All trials | (Intercept) | −3.06 | [−7.15, 0.93] | 93.45 | 17,486 | All trials | (Intercept) | 1.02 | [−4.42, 6.48] | 64.35 | 23,656 |
| | I | 0.56 | [0.41, 0.71]* | 100.00 | 23,950 | | I | 0.58 | [0.37, 0.79]* | 100.00 | 30,658 |
| | H | 1.09 | [−1.23, 3.44] | 82.45 | 18,259 | | H | −1.68 | [−4.85, 1.49] | 84.90 | 24,213 |
| | DKL | −13.13 | [−21.48,−4.84]* | 99.90 | 20,134 | | DKL | 0.49 | [−10.83, 11.90] | 53.39 | 29,994 |
| | Baseline | −0.4 | [-0.42,−0.38]* | 100.00 | 24,410 | | Baseline | −0.71 | [−0.74,−0.68]* | 100.00 | 35,733 |
| | RT | 0.02 | [−0.17, 0.21] | 100.00 | 23,370 | | RT | 0.17 | [−0.09, 0.42] | 89.89 | 31,833 |
| | Parameter | Median | 95% CI | pd (%) | ESS | | Parameter | Median | 95% CI | pd (%) | ESS |
| Correct trials | (Intercept) | −2.39 | [−7.43, 2.67] | 82.29 | 25,551 | Correct trials | (Intercept) | −2.79 | [−9.75, 4.31] | 78.18 | 25,261 |
| | I | 0.57 | [0.21, 0.93]* | 99.91 | 31,617 | | I | 0.34 | [−0.16, 0.83] | 90.98 | 29,758 |
| | H | 0.56 | [−2.42, 3.49] | 64.42 | 26,985 | | H | 0.64 | [−3.51, 4.71] | 61.97 | 25,637 |
| | DKL | −11.97 | [−22.91,−1.20]* | 98.41 | 30,657 | | DKL | −6.54 | [−21.36, 8.73] | 79.68 | 31,266 |
| | Baseline | −0.41 | [-0.43,−0.38]* | 100.00 | 36,828 | | Baseline | −0.68 | [−0.72,−0.64]* | 100.00 | 33,299 |
| | RT | 0.07 | [−0.17, 0.30] | 71.85 | 32,381 | | RT | 0.24 | [−0.08, 0.57] | 92.58 | 30,517 |
| | Parameter | Median | 95% CI | pd (%) | ESS | | Parameter | Median | 95% CI | pd (%) | ESS |
| Error trials | (Intercept) | −0.79 | [−8.01, 6.16] | 58.84 | 20,239 | Error trials | (Intercept) | 8.23 | [−0.33, 16.79] | 96.97 | 26,770 |
| | I | −0.04 | [−0.27, 0.19] | 63.10 | 28,734 | | I | 0.27 | [−0.01, 0.56] | 97.01 | 32,348 |
| | H | 1.06 | [−2.87, 5.13] | 69.95 | 21,448 | | H | −5.15 | [−9.99,−0.28]* | 98.09 | 26,886 |
| | DKL | −18.52 | [−31.55,−5.56]* | 99.71 | 26,800 | | DKL | 11.38 | [−4.68, 27.60] | 91.88 | 31,452 |
| | Baseline | −0.41 | [-0.44,−0.37]* | 100.00 | 28,102 | | Baseline | −0.79 | [-0.83,−0.74]* | 100.00 | 30,943 |
| | RT | −0.24 | [−0.56, 0.07] | 93.27 | 28,149 | | RT | −0.02 | [−0.40, 0.38] | 52.89 | 33,367 |

The online version of this article includes the following source data for table 3:

**Source data 1.** Processed data input into the linear mixed modeling analysis for the cue-target 2AFC task.

significant variance in the late time window in the linear mixed model analysis for error trials. This discrepancy suggests that the predictive power of surprise may be due to variance it shares with entropy for the error trials. As expected, the pre-feedback baseline pupil dilation explained a significant amount of variance in the feedback-locked pupil response in both time windows for the correct and error trials (*Table 3*). RT did not explain significant variance in either time window for the correct or error trials.

## Ideal learner model fits for the letter-color 2AFC task: correlation analysis

We first confirmed that the odd-ball task yielded a gradient of probability distributions according to the task as designed: the 33%, 50%, and 84% stimulus-pair frequency conditions had mean final probabilities of 0.034 (SD = 0.001), 0.050 (SD = 0.003), and 0.112 (SD = 0.004), respectively. For each participant, the final probabilities of each letter-color pair at the end of the odd-ball task corresponded to the prior distribution and were entered into the ideal learner model for the letter-color 2AFC task.

For the letter-color 2AFC task, the three information-theoretic variables are shown as a function of task trial in *Figure 3G* and as a function of stimulus-pair frequency in *Figure 3H*. Multicollinearity was again assessed by the correlations between the model parameters (information gain vs. surprise: $r = 0.33$, $p < 0.001$; information gain vs. entropy: $r = -0.51$, $p < 0.001$; surprise vs. entropy: $r = 0.05$,

p < 0.001). The trial-by-trial correlation of the post-feedback pupil response with the model parameters was repeated for the letter-color 2AFC task. The time course of the resulting correlation coefficients showed a pattern in which the pupil scaled positively with information gain shortly (~0.5 s) after feedback onset until about 1.75 s into the feedback interval spanning across the early time window (*Figure 3I*, purple line). The post-feedback pupil response also scaled negatively with entropy across the same interval as for information gain (*Figure 3I*, yellow line). No scaling between the post-feedback pupil response and surprise was obtained.

We repeated the correlation analysis of the information-theoretic variables to the feedback-locked pupil response separately for error and correct trials. The post-feedback pupil response scaled positively with information gain across the early time window for both error and correct trials (*Figure 3J*, red and blue lines). While no scaling between the post-feedback pupil response with surprise was obtained for error or correct trials (*Figure 3K*), the relationship with information gain was mirrored in a negative scaling with entropy for correct trials only (*Figure 3L*).

**Table 4.** Linear mixed model results for the letter-color 2AFC task.

Explanation of abbreviations, rows: *I*, Shannon surprise predictor variable; *H*, entropy; $D_{KL}$, information gain; *Baseline*, pre-feedback baseline pupil dilation; *RT*, reaction times. Columns: *95% CI*, the 95% credible interval of the median posterior distribution; *pd*, the probability (in percentage) of direction; *ESS*, effective sample size; *indicates strong evidence that the parameter has a positive/negative effect on the post-feedback pupil response.

| Letter-color 2AFC task | | | | | | | | | | |
|---|---|---|---|---|---|---|---|---|---|---|
| Early time window | | | | | | Late time window | | | | |
| | Parameter | Median | 95% CI | pd (%) | ESS | | Parameter | Median | 95% CI | pd (%) | ESS |
| All trials | (Intercept) | 22.1 | [13.21, 30.86]* | 100.00 | 26,356 | All trials | (Intercept) | –3.06 | [–15.23, 9.09] | 69.53 | 22,915 |
| | I | –0.02 | [–0.13, 0.09] | 64.55 | 34,055 | | I | 0.09 | [–0.06, 0.24] | 87.28 | 26,755 |
| | H | –4.5 | [-6.40,–2.58]* | 100.00 | 25,959 | | H | 0.59 | [–2.03, 3.23] | 67.52 | 23,997 |
| | DKL | 60.07 | [39.51, 80.87]* | 100.00 | 26,249 | | DKL | –24.32 | [–53.07, 4.56] | 95.10 | 33,991 |
| | Baseline | –0.26 | [-0.27,–0.24]* | 100.00 | 44,190 | | Baseline | –0.58 | [–0.60,–0.56]* | 100.00 | 26,963 |
| | RT | 1.46 | [1.03, 1.90]* | 100.00 | 47,747 | | RT | –0.16 | [–0.75, 0.42] | 70.89 | 30,625 |
| | Parameter | Median | 95% CI | pd (%) | ESS | | Parameter | Median | 95% CI | pd (%) | ESS |
| Correct trials | (Intercept) | 14.94 | [4.81, 25.11]* | 99.80 | 24,140 | Correct trials | (Intercept) | –4.35 | [–18.46, 9.70] | 72.67 | 19,889 |
| | I | 5.15e-03 | [–0.12, 0.13] | 53.31 | 30,199 | | I | 0.14 | [–0.04, 0.31] | 93.99 | 21,789 |
| | H | –3.05 | [-5.25,–0.87]* | 99.72 | 23,801 | | H | 0.8 | [–2.24, 3.85] | 69.60 | 19,607 |
| | DKL | 50.58 | [24.08, 77.54]* | 100.00 | 25,756 | | DKL | –32.31 | [–69.66, 5.65] | 95.47 | 19,778 |
| | Baseline | –2.6 | [-0.27,–0.24]* | 100.00 | 37,674 | | Baseline | –0.57 | [–0.58,–0.55]* | 100.00 | 27,486 |
| | RT | 1.36 | [0.86, 1.87]* | 100.00 | 28,719 | | RT | –0.2 | [–0.90, 0.49] | 72.05 | 23,635 |
| | Parameter | Median | 95% CI | pd (%) | ESS | | Parameter | Median | 95% CI | pd (%) | ESS |
| Error trials | (Intercept) | 15.37 | [–2.50, 33.00] | 95.54 | 27,183 | Error trials | (Intercept) | –11.89 | [–35.13, 10.58] | 84.82 | 23,997 |
| | I | –0.22 | [–0.46, 0.02] | 96.50 | 32,906 | | I | –0.14 | [–0.47, 0.17] | 81.38 | 28,603 |
| | H | –1.84 | [–5.71, 2.03] | 82.42 | 26,618 | | H | 3.12 | [–1.79, 8.22] | 89.03 | 23,597 |
| | DKL | 43.74 | [11.29, 76.03]* | 99.63 | 25,265 | | DKL | –22.82 | [–65.81, 20.46] | 85.06 | 25,965 |
| | Baseline | –0.29 | [-0.32,–0.26]* | 100.00 | 38,565 | | Baseline | –0.68 | [–0.72,–0.65]* | 100.00 | 32,302 |
| | RT | –0.59 | [–1.44, 0.25] | 91.18 | 30,075 | | RT | –0.65 | [–1.79, 0.48] | 87.33 | 31,779 |

The online version of this article includes the following source data for table 4:

**Source data 1.** Processed data input into the linear mixed modeling analysis for the letter-color 2AFC task.

## Ideal learner model fits for the letter-color 2AFC task: linear mixed model analysis

To assess the unique variance explained by each of the model parameters during the letter-color 2AFC task, we compared two linear mixed models with the feedback-locked pupil response as the dependent variable (see Materials and methods). The model comparison was performed independently for the early and late time windows. For the letter-color 2AFC task, Model 1 performed slightly better than Model 2 for both the early ($R^2$ = 0.16, σ = 8.5) and late time window ($R^2$ = 0.31, σ = 11.68; see *Appendix 2—table 2* for the model comparison results), indicating that the inclusion of an interaction term between information gain and entropy did not lead to a better model fit of the post-feedback pupil response. The parameters of Model 1 are presented in *Table 4*. The linear mixed model analysis corroborated the correlation analysis: in the early time window, the predictor variables of entropy and information gain explained significant variance in the feedback-locked pupil response; in the late time window, in contrast, none of the information-theoretic predictor variables were significant. As expected, the pre-feedback baseline pupil dilation explained a significant amount of variance in the feedback-locked pupil response in both time windows. In addition to the pre-feedback baseline pupil, reaction time was a significant predictor of the post-feedback pupil response in the early time window of the letter-color 2AFC task.

When examining the correct and error trials separately, Model 2 never outperformed Model 1 but was sometimes significantly worse (*Appendix 2—table 2*); therefore, we only report the parameter estimates for Model 1 (*Table 4*). The linear mixed modeling results again corroborated the correlation analysis. For the correct trials in the early time window ($R^2$ = 0.16, σ = 8.61), showed that both the predictor variables of entropy and information explained significant variance in the feedback-locked pupil response (and scaled in opposite directions); for correct trials in the late time window, ($R^2$ = 0.29, σ = 11.96), none of the information-theoretic predictor variables were significant. The linear mixed modeling results for the error trials in the early time window ($R^2$ = 0.24, σ = 7.51) showed that the predictor variable of information gain explained significant variance in the feedback-locked pupil response, while none of the information-theoretic predictor variables were significant in the late time window ($R^2$ = 0.42, σ = 10.11). Again, as expected, the pre-feedback baseline pupil dilation explained a significant amount of variance in the feedback-locked pupil response in both time windows for the correct and error trials (*Table 3*). In addition to the pre-feedback baseline pupil, reaction time was a significant predictor of the post-feedback pupil response for the correct trials in the early time window of the letter-color 2AFC task.

## Discussion

In the current study, we investigated whether the pupil's response to decision outcome (i.e., feedback) in the context of associative learning reflects a prediction error defined operationally as an interaction between stimulus-pair frequency and accuracy. Thereafter, we tested whether these prediction error signals correlated with information gain, defined formally as the KL divergence between posterior and prior belief distributions of the ideal observer. We also explored how prediction error signals changed over time (3 s) with respect to the trial-wise feedback on decision outcome across two independent associative learning paradigms. Information-theoretic variables were derived from an ideal learner model and fit to the post-feedback pupil response at the trial-by-trial level. For completeness, we computed Shannon surprise and entropy and examined their relationship with the post-feedback pupil response.

Results showed that signed prediction error signals, which illustrate the relationship between frequency and accuracy, were evident in distinct time windows: during the later time window for the cue-target 2AFC task and in the early window for the letter-color 2AFC task. The post-feedback pupil response correlated with information gain in the early time window across both tasks, while the direction of this scaling (qualitatively) differed per task. Shannon surprise was associated with the later time window in the cue-target 2AFC task, while entropy (in addition to information gain) related to the early time window in the letter-color 2AFC task. Our findings offer novel insights into the relationship between prediction error signals in post-feedback pupil responses and information processing by investigating how information-theoretic variables reveal the underlying computational processes driving the interactions between stimulus-pair frequency and accuracy.

## Information gain was reflected in the early post-feedback pupil response

The results supported our hypothesis that prediction error signals in the post-feedback pupil dilation reflected information gain, indicating that the pupil's response to decision outcome reflects the amount of information gained during associative learning. Across both task contexts, the post-feedback pupil response correlated with information gain within the early (0.75–1.25 s) but not late time window (2.5–3 s) (see *Figure 3C and I*). This early scaling between the pupil response following feedback onset and information gain suggests that the difference between the posterior and prior belief distributions is transiently reflected in pupil dilation shortly after observing the decision outcome about the stimulus pairs.

For the first time, we show that the direction of the relationship between post-feedback pupil dilation and information gain (defined as the KL divergence between posterior and prior belief distributions) was context dependent. Specifically, in the cue-target 2AFC task, there was a negative effect of information gain on pupil dilation: the pupil response was smaller for larger values of information gain. In contrast, in the letter-color 2AFC task, there was a positive effect of information gain on pupil dilation: the pupil response was larger for larger values of information gain. This pattern of results was apparent in the correlation analysis across the pupil time course as well as in the linear mixed model analysis accounting for shared variance from other explanatory variables on the time windows of interest.

The entropy as a function of task trial differed between these task contexts (compare *Figure 3A* right panel with *Figure 3G* right panel): At the end of the odd-ball task, participants were exposed to the letter-color pairs in the high-frequency (84%) more often as compared with the lower-frequency conditions (33% and 50%). Therefore, stronger expectations about letter-color pairs for the 84% letter-color condition are represented by larger priors at the start of the letter-color 2AFC task. This increasing entropy in the letter-color 2AFC task can be attributed to the fact that the letter-color pair conditions are balanced in terms of frequency of presentation while the prior distribution was not uniform. In other words, there is increasing average uncertainty driven by the stronger prior expectations in the subsequent fully balanced letter-color 2AFC task. To verify that the direction of entropy depended on the prior distribution chosen, we ran the ideal learning model on the letter-color 2AFC data using a uniform prior distribution (see *Figure 3—figure supplement 1*). Although it is tempting to speculate that the direction of the relationship between pupil dilation and information gain may be due to either increasing or decreasing entropy as the task progressed, we must refrain from this conclusion. We note that the two tasks differ substantially in terms of design with other confounding variables and therefore cannot be directly compared to one another. We expand on these limitations in the section below (see Limitations and future research).

The results from the cue-target 2AFC task are in line with those reported by *O'Reilly et al., 2013* in which pupil dilation was also smaller for larger values of information gain centered around 1 s following target onset in a saccadic planning task. Although we did not explicitly dissociate surprise and information gain in the cue-target 2AFC task design, as did O'Reilly et al., we found converging results related to the pupil's response to information gain (or 'updating'). While O'Reilly et al. also found a positive relationship between pupil dilation and surprise, this surprise effect in their saccadic planning task emerged earlier than the scaling with information gain, unlike in the cue-target 2AFC task here. In the cue-target 2AFC task, we furthermore see a sustained surprise effect that spans most of the post-feedback interval, while the surprise effect in the previous saccadic planning task was transient. The discrepant results further illustrate the importance of task context for interpreting the relationship between pupil dilation and information processes. For instance, saccadic RTs were reported to scale with target expectancy; however, in the current study, we are investigating the post-feedback intervals that did not require any motor responses from the participants. Furthermore, while we found early scaling between the post-feedback pupil response and information gain in both the cue-target and letter-color 2AFC tasks, the presence of a relationship between the post-feedback pupil response with surprise and entropy differed across tasks (compare *Figure 3C with I*).

*O'Reilly et al., 2013* suggested that 'pupil increases during learning are driven by uncertainty, or the influence of uncertainty on learning, rather than by learning or change per se'. The results taken together across these two associative learning paradigms are in line with their suggestion. The results of the current study are also in line with the proposition of *Zénon, 2019* that 'the ensemble of

phenomena that trigger changes in pupil-linked arousal all depend on a basic underlying information theoretic process: the update of the brain internal models'. Furthermore, from the contrast of the two associative learning tasks presented here, we can confirm that the pupil does not respond simply to (Shannon) surprise, because it does not always follow the frequency of occurrence of the stimulus pairs, independently of the task. Instead, the pupil seems to respond to the amount of information provided by stimuli about the task variables. Other studies have shown that the relationship between information gain and surprise reflected in pupil dilation is context dependent. One study testing children found that pupil dilation positively correlated with information gain, but not surprise, only when children were actively making predictions about water displacement, but not when they evaluated outcomes about water displacement without having to make predictions (*Colantonio et al., 2023*). While *Shirama et al., 2024* showed that the covariance of trial-wise pupil dilation and information gain depended on individual differences in accuracy, pupil dilation did not reflect surprise during change points in their study. *Fleischmann et al., 2025* reported that pupil dilation correlated with both surprise and information gain, but information gain was the more accurate predictor. Zénon discusses the negative scaling of pupil dilation with information gain reported by O'Reilly et al. as contradicting the hypothesis that the pupil dilation will increase in proportion to how much novel sensory evidence is used to update current beliefs. Here, we provide additional evidence that the direction of this relationship between pupil dilation and information gain needs more context.

Interestingly, only in the letter-color 2AFC task, results showed that the post-feedback pupil response negatively correlated with entropy in the early but not late time window, mirroring the positive scaling between the pupil response and information gain (see *Figure 3I*). While volatility and entropy describe different phenomena, they are interconnected through their relationship with uncertainty and predictability. High volatility typically corresponds to higher entropy, reflecting a system's complexity and unpredictability (*Sheraz et al., 2015*). Previous work has shown that both tonic and phasic fluctuations in pupil dilation may track volatility in the environment, sometimes referred to as unexpected uncertainty (*Nassar et al., 2012*; *Vincent et al., 2019*; *Filipowicz et al., 2020*; *Harris et al., 2022*; *Pajkossy et al., 2023*; *Murphy et al., 2021*). Furthermore, noradrenaline plays a crucial role in how organisms track and respond to volatility in their environments, as it can signal when to update beliefs and expectations, enhancing the brain's ability to adapt to fluctuations (*Sales et al., 2019*; *Nassar, 2024*). Using a probabilistic reversal learning task, *Pajkossy et al., 2023* reported that state entropy positively predicted post-feedback pupil size changes and interacted with the reversal probability of stimulus-reward contingencies across three different variations of the experiment. The scaling on entropy with the post-feedback pupil response reported by Pajkossy et al. occurred overall later in time with respect to feedback onset (ranging from feedback onset to 6 s depending on experimental variation) as compared with the letter-color 2AFC task. While more research is needed to understand these discrepant results, certainly differences between the reversal learning tasks in Pajkossy et al. and 'simple' associative learning required in the letter-color 2AFC task could play a role, such as the presence or lack of changing stimulus probabilities and the inclusion of a point-based reward system for feedback.

## The relationship between signed prediction error signals and information gain

As the term "ideal learner" suggests, the information-theoretic variables are computed based on the stimulus events shown to participants but are not sensitive to their actual behavioral performance. Of course, participants do not always act as ideal learners and make errors of observation, inference, and motor responses. Therefore, as a complementary analysis, we sought to examine the relationship between the performance accuracy and the information-theoretic variables to understand which computational processes may be underlying the stimulus-pair frequency and accuracy interactions reflected in the post-feedback pupil response.

In the cue-target 2AFC task, a prediction error should occur when the target orientation did not match the expected orientation based on the learned contingencies. A signed prediction error signal was obtained in the late time window, with the low-frequency (20%) error trials driving the interaction effect (see *Figure 1F*). Converging with this late signed prediction error signal, the correlations between surprise and the post-feedback pupil response differed for the error as compared with correct trials in the late time window (see *Figure 3E*). Specifically, the post-feedback pupil response

during error trials showed larger correlation coefficients with surprise as compared with correct trials from about 1.75 to 3 s following the target onset. Thus, a signed prediction error signal defined by the interaction between frequency and accuracy in the late time window task seems to be driven by *surprise* and not information gain for the cue-target 2AFC. In line with this result, *Lavín et al., 2013* also found a surprise-driven effect in pupil dilation specifically following negative feedback presentation during a learning gambling task. This surprise-driven effect was evident in early (~500 ms) and later time windows (1200–1300 and 1700–2400 ms following feedback onset). The direction of this surprise-driven effect could be interpreted in relation to sensory evidence. For instance, *Colizoli et al., 2018* measured pupil dilation during a random dot discrimination paradigm with hard and easy levels of motion coherence that did not involve probabilistic learning. The signed prediction error signal in pupil dilation was found to depend on sensory evidence within a late time window (3–6 s) following feedback, consistent with findings from the cue-target 2AFC task (see *Figure 1D and F*). The relationship between surprise and prediction error signal is partially in line with reward-linked feedback signals in *Van Slooten et al., 2018*. Using a probabilistic value-based reinforcement learning task, van Slooten et al. reported that the early post-feedback pupil response (<~2 s) was modulated by uncertainty about the value of options (with smaller differences between value options resulting in larger pupil dilation) but was not affected by violations of value beliefs (i.e., surprise). In contrast, the later post-feedback pupil response (around 2–3 s) positively reflected the degree to which outcomes violated current value beliefs. However, the direction of the late prediction error signal indicated that worse-than-expected outcomes were related to smaller pupil sizes, which seems to be at odds with other work showing that pupils generally dilate more when performance is worse than expected such as during errors (*Colizoli et al., 2018*; *de Gee et al., 2021*; *Urai et al., 2017*; *Braem et al., 2015*; *Critchley et al., 2005*; *Maier et al., 2019*; *Murphy et al., 2016*; *Rondeel et al., 2015*; *Wessel et al., 2011*; however, see also *Lavín et al., 2013*). The authors speculated that the late reward prediction error signal may reflect the firing pattern of phasic dopamine neurons, and other work supports the notion of a significant component of dopamine signaling being reflected in pupil dilation (*Colizoli et al., 2018*; *Lloyd et al., 2023*; *de Gee et al., 2014*). A key difference between the current study and van Slooten et al. is the absence of reward-driven feedback during associative learning in the current study.

In the letter-color 2AFC task, a prediction error should occur when the participant expected that the letter-color pair did 'match', but in fact they did not match, or vice versa. A signed prediction error signal was significant in the early time window (see *Figure 2E*). The direction of this interaction effect indicated that the pupil response difference between errors and correct trials increased as letter-color pair frequency increased (see *Figure 2E*). We note that the direction of the signed prediction error might seem counterintuitive as it relates to information gain, because stronger predictions (i.e. higher-frequency observations) often result in less information gain following outcome observation. As discussed above, in the letter-color 2AFC task, the amounts of surprise and information gain are highest for the high-frequency as compared with the lower-frequency conditions related to the increasing entropy across trials (see *Figure 3H*, left-hand and middle panels; compare with *Figure 3—figure supplement 1*). Converging with this early signed prediction error signal, correlations between both information gain and entropy with the post-feedback pupil response were obtained in the early time window (see *Figure 3I*); however, no differences between correlations on error as compared with correct trials were obtained for either information-theoretic variable (see *Figure 3J and L*). Understanding the information processing in relation to performance accuracy may be crucial for disentangling the early signed prediction error signal. An alternative contributing factor that we did not explicitly test for is the participants' confidence about the stimulus-pair associations. Using an orientation discrimination task, *de Gee et al., 2021* reported that the early post-feedback pupil response was largest on error trials and smallest on correct trials when participants were most (subjectively) confident about their choices. Although we did not ask participants to report on their confidence about choices made, both RT and the strength of the stimulus-pair priors could be taken as a proxy for confidence (*Urai et al., 2017*; *Sanders et al., 2016*). In line with this, we did obtain interactions between stimulus-pair frequency and accuracy in both RT and the early post-feedback pupil response.

In sum, since the ideal learner model does not capture participant errors, we aimed to connect these two approaches of analyzing prediction errors by fitting the information-theoretic variables to the pupil response during error and correct trials independently. Signed prediction error signals

defined by the interaction between frequency and accuracy were observed in the late time window for the cue-target 2AFC task and in the early window for the letter-color 2AFC task. Shannon surprise was related to the later component in the cue-target 2AFC task, while both information gain and entropy were related to the early component in the letter-color 2AFC task.

## Limitations and future research

This study has some limitations. First, the two associative learning paradigms differed in many ways and were not directly comparable. For instance, the shape of the mean pupil response function differed across the two tasks in accordance with a visual or auditory feedback stimulus, and it is unclear whether these overall response differences contributed to any differences obtained between task conditions within each task. We are unable to rule out whether so-called 'low level' effects such as the initial constriction to visual stimuli in the cue-target 2AFC task as compared with the dilation in response to auditory stimuli in letter-color 2AFC task could confound correlations with information gain. Future work should strive to disentangle how the specific aspects of the associative learning paradigms relate to prediction errors in pupil dilation by systematically manipulating design elements within each task. Task context clearly determines the relationship between the post-feedback pupil response and the information-theoretic variables, as it determines the uncertainty conditions surrounding decision-making. To determine exactly how the different associative learning tasks relate to different temporal components of model updating is beyond the scope of the current study, but we speculate that hybrid predictive coding models may be able to account for fast (bottom-up) and slow (top-down) prediction errors reflected in pupil dilation (*Tscshantz et al., 2023*). Second, we did not design the associative learning paradigms to orthogonalize the information-theoretic variables, such as was done in O'Reilly et al. Indeed, some multicollinearity was evident between the information-theoretic variables in each of the two tasks. Cleverer associative learning paradigms may be able to overcome this limitation. Third, we are unable to attribute the relationship between computational variables and pupil dilation to specific neural mechanisms or neuromodulatory systems with the current study. Previous work has shown how neuromodulatory systems relate to learning and decision-making under uncertainty and the ability of the pupil to reflect these underlying computational processes (*de Gee et al., 2017*; *Murphy et al., 2014a*; *Nassar et al., 2012*; *Murphy et al., 2014b*; *Vincent et al., 2019*; *Filipowicz et al., 2020*; *Harris et al., 2022*; *Pajkossy et al., 2023*; *Murphy et al., 2021*; *Meyniel, 2020*). Future research should aim at identifying the neural mechanisms involved in the processes underlying associative learning as reflected in pupil dilation across all phases of a decision process, ideally through computational theory. Finally, while we acknowledge the potential relevance of subjective factors, such as the participants' overt confidence reports, in understanding prediction errors and pupil responses, the current study focused on the more objective, model-driven measure of information-theoretic variables. This approach aligns with our use of the ideal learner model, which estimates information-theoretic variables while being agnostic about the observer's subjective experience itself. Future research is needed to explore the relationship between information-gain signals in pupil dilation and the observer's reported experience of or awareness about confidence in their decisions.

Understanding prediction errors through pupil dilation within an information theory framework can illuminate predictive processing mechanisms in several significant ways. Pupil dilation acts as a physiological marker of cognitive and emotional responses, allowing researchers to quantitatively assess how discrepancies between expected and actual outcomes impact cognitive processing (*Browning et al., 2015*; *Pajkossy et al., 2023*; *Stemerding et al., 2022*). A framework for interpreting pupil dilation in terms of information theory may also enable exploration of how prediction errors function at various levels of a hierarchical model, helping researchers examine the interaction between high-level expectations and lower-level sensory inputs across different timescales, which informs the overall predictive model (*Iglesias et al., 2013*; *Mathys et al., 2011*; *Dijkstra et al., 2025*). Analyzing pupil responses in relation to prediction errors can reveal the extent of new information being processed and its influence on future predictions, thereby enhancing our understanding of learning and adaptation dynamics by elucidating feedback mechanisms in predictive processing and demonstrating how the brain adjusts its predictions based on new information. Given that pupil dilation is a peripheral marker of the brain's central arousal states, understanding its relationship with prediction errors can help disentangle the cognitive and affective components of predictive processing, providing a more comprehensive view of how the brain navigates uncertainty (*Nassar, 2024*; *Pulcu and Browning,*

*2019*). By integrating such insights, researchers can gain a deeper understanding of the mechanisms underlying predictive processing and how the brain continuously updates its internal models based on new experiences.

## Conclusion

To conclude, the results provide evidence for Zénon's general assumption that pupil dilation can be described by an information-theoretic perspective. Clearly, task context plays a key role in the relationship between the information-theoretic variables and the post-feedback pupil response, as may be expected. The temporal dynamics of these prediction error signals should be carefully considered, as certain components tended to emerge around the peak of the canonical impulse response function and others may be sustained over time. These subtleties highlight the importance of adopting a model-based approach for characterizing the computational processes driving prediction errors as reflected in pupil dilation. Taken together, the post-feedback pupil response is a complex and multifaceted signal that reflects different components of information processing during associative learning. The physiological response of the pupil provides a unique window into the brain's computations involved in model updating. More work is needed to link the information-theoretic variables reflected in the post-feedback pupil response with their underlying neuromodulatory mechanisms (*Grujic et al., 2024*).

## Materials and methods
### Data sets: decision-making tasks in associative learning paradigms

We analyzed two data sets related to associative learning in which pupil dilation was recorded during decision making; one of these data sets was publicly available (*Rutar et al., 2023*), the other was collected for the study purposes. In the decision-making tasks of both data sets, participants had to make a two-alternative forced choice (2AFC) on each trial based on visual and/or auditory information. For the current analyses, the dependent variables of interest were participants' accuracy, reaction time (RT), and the feedback-locked pupil response. In the first data set (see Materials and methods section 'Data set #1: Cue-target 2AFC task' for further details), participants learned probabilistic contingencies between stimuli during the 2AFC decision-making task itself. In the second data set (see Materials and methods section 'Data set #2: Letter-color 2AFC task' for further details), there was an implicit learning phase prior to the 2AFC decision-making task during which the participants completed an odd-ball detection task that was irrelevant to the probabilistic contingencies between pairs of stimuli being presented. Pupil dilation was continuously measured in both data sets during each of the 2AFC decision-making tasks. The data sets consisted of independent samples of participants.

These data sets were chosen to analyze because we were able to quantify the post-feedback pupil dilation as the interaction between stimulus-pair frequency and accuracy as well as adapt an information-theoretic model of trial-by-trial learning in both task paradigms. The pupil is known to scale positively with errors as compared with correct responses (*Colizoli et al., 2018*; *de Gee et al., 2021*; *Urai et al., 2017*; *Braem et al., 2015*; *Critchley et al., 2005*; *Maier et al., 2019*; *Murphy et al., 2016*; *Rondeel et al., 2015*; *Wessel et al., 2011*). In addition, studies have shown that pupil dilation positively scales with unexpected as compared with expected events (*de Gee et al., 2021*; *O'Reilly et al., 2013*; *Preuschoff et al., 2011*; *Alamia et al., 2019*; *Bianco et al., 2020*; *Friedman et al., 1973*; *Kamp and Donchin, 2015*; *Kloosterman et al., 2015*; *Knapen et al., 2016*; *Kuchinke et al., 2007*; *Lavín et al., 2013*; *Liao et al., 2016*; *Qiyuan et al., 1985*; *Raisig et al., 2010*; *Silvestrin et al., 2021*; *Wetzel et al., 2016*; *Zhao et al., 2019*; *Ghilardi et al., 2024*). Prediction errors can be operationalized as a function of task conditions related to stimulus expectations. For instance, a main effect of stimulus frequency reflects an unsigned prediction error, since the different frequency conditions correspond to different levels of expectancy (in the simplest form, a contrast of unexpected vs. expected). A main effect of accuracy (categorized post-hoc based on task performance) indicates an error signal about the binary outcome of a decision (correct vs. incorrect). However, an error signal is not the same as a prediction error signal, because an error alone does not necessarily convey information regarding expectation. In other words, errors on accuracy do not contain quantitative information regarding a difference between what was expected and what occurred. Expectations can modulate a main effect of accuracy in the post-feedback pupil response, indicating whether the outcome of the

participant's accuracy (correct or incorrect) is better or worse than expected (*Colizoli et al., 2018*; *de Gee et al., 2021*). A signed prediction error in the context of associative learning would therefore be evidenced by an interaction between stimulus-pair frequency and accuracy.

In the current data sets, we were interested in comparing the condition when a participant makes an error, and the outcome was expected, with the condition in which they make an error and the outcome was unexpected (and similarly for when they are correct about their decision). If the signed prediction error signal correlates with information gain, then we would expect a larger difference between error and correct trials for the condition with weaker expectations as compared with stronger expectations. In other words, more information is gained from a decision outcome in conditions with more uncertainty as compared with less uncertainty. Finally, we explored whether a definition of prediction error as an interaction between stimulus-pair frequency and accuracy will correlate with information gain.

## Data set #1: cue-target 2AFC task

Independent analyses that focused on the relationship between the participants' pupil responses and Bayesian learning mechanisms have been previously published based on the same data set (*Rutar et al., 2023*). The data are publicly available and have been re-analyzed (including pre-processing) in the current paper to answer a complementary but conceptually distinct research question compared with the (*Rutar et al., 2023*) paper. The relevant methods that have been previously published are summarized here.

### Participants and informed consent

From the thirty participants included in the published data set, six participants missed responses in at least one of the conditions required for the main two-way repeated-measures ANOVA and were therefore excluded from the statistical analysis. The final sample consisted of 24 participants aged 19–42 years ($M$ = 23.3, SD = 4.9, 18 women). All participants gave written informed consent before participating and were compensated for participation.

### Task and procedure

Participants performed a 2AFC decision-making task on the expected orientation direction (left vs. right) of the target stimulus (Gabor patches; spatial frequency = 0.033, opacity = 0.5, 400 x 400 pixels) while pupil dilation was recorded (*Figure 1A*). A chin rest was used to keep the distance (50 cm) to the computer screen constant (1920 x 1,080 pixels, 120 Hz). This setup resulted in a Gabor patch with a visual angle of 12.1°. Participants were instructed to use the visual and auditory cues to predict the orientation of the upcoming target stimulus in each trial and to respond (by left or right button press corresponding to a left or right prediction, respectively) as soon as they knew which target orientation would appear. Participants were instructed to wait until cue offset to make their predictions. The orientation of the target was probabilistically determined by the combination of the preceding visual and auditory cues. The cue-target contingencies were not communicated in advance to the participants. The cue-target mappings were counterbalanced between participants in such a way that half of the participants saw the square followed by a right-oriented grating and a diamond followed by a left-oriented grating in 80% of the trials, and for the other half of the participants, this mapping was reversed (i.e. square –> left and diamond –> right). In the remaining 20% of the trials, the participants received the reversed cue-target mapping with respect to their 80% mapping condition. On half of the trials, an auditory tone ("C" octave 4; 300ms) was presented together with the onset of the visual cue. In the first half of the experiment (phase 1), this auditory tone was uninformative of the upcoming target orientation and could essentially be ignored. After phase 1 was completed, participants took a short break and were informed that the cue-target contingency rule would change; for the purposes of the current analysis, we only inspected phase 1 of the original experiment. Phase 1 consisted of 200 trials per participant. The order of the trials was randomly presented to each participant. The entire session took about 1.5 hr to complete (~45 min for phase 1). Stimuli were isoluminant and had a gray background.

### Trial structure

Each trial of the cue-target task consisted of a fixation period (1.5 s), a cue period (1 s), a decision period during which the participant responded by pressing either the left or right button (RT), a delay

period (3 s), and the target period (3 s) which served as feedback for the cue-target contingencies. A vertically oriented Gabor patch was presented on screen except for the target period. Participants were instructed to keep their gaze centered at the fixation cross in the middle of the screen. The fixation cross remained on screen except for the cue period, during which either a square or diamond appeared. During the cue period, a square or diamond would indicate the upcoming orientation direction of the target stimulus with a certain probability (20% vs. 80%). Note that in the cue-target task, the target period served as trial-by-trial feedback on the accuracy of the participants' cue-target predictions.

## Data acquisition and preprocessing

Pupil dilation of the right eye was continuously recorded during phase 1 of the cue-target task using an SMI RED500 eye-tracker (SensoMotoric Instruments, Teltow/Berlin, Germany). The sampling rate was 500 Hz. Using custom Python code, the following steps were applied to the entire pupil dilation time series: (i) linear interpolation around missing samples (0.15 s before and after each missing event), (ii) linear interpolation around blinks or saccade events based on spikes in the temporal derivative (0.15 s before and after each nuisance event; note that blinks and saccades were considered as a single nuisance event), (iii) band-pass filtering (third-order Butterworth, 0.01–6 Hz), (iv) responses to nuisance events were removed using linear regression (nuisance responses were estimated using deconvolution; *Knapen et al., 2016*) and (v) the residuals of the nuisance regression were converted to percent signal change with respect to the temporal mean. The size of the interpolation window preceding nuisance events was based on previous literature (*Urai et al., 2017*; *Knapen et al., 2016*; *Winn et al., 2018*). After interpolation based on data markers and/or missing values, remaining blinks and saccades were estimated by testing the first derivative of the pupil dilation time series against a threshold rate of change. The threshold for identifying peaks in the temporal derivative is data-driven, partially based on past work (*Colizoli et al., 2018*; *de Gee et al., 2017*; *Rutar et al., 2023*). The output of each participant's pre-processing pipeline was checked visually. Once an appropriate threshold was established at the group level, it remained the same for all participants (minimum peak height of 10 units). Trials that were faster or longer than 3 times the standard deviation of the $Z$-transformed RT distribution of each participant were excluded from the analysis, because there was no maximum response window (1.5% of the total number of trials were excluded; *Berger and Kiefer, 2021*).

## Differences with *Rutar et al., 2023*

The main differences between the current work and *Rutar et al., 2023* are the following. First, in the current analysis, we are only considering the first 200 trials (referred to as 'phase 1') out of the 400 trials in total. After 200 trials (referred to as 'phase 2'), the cue-target contingencies switched according to a specific rule. The change in rule-based contingencies prevented us from applying the ideal learner model to both phases of the task. Crucially, trials in phase 1 were independent from trials in phase 2 as they took place earlier in time, and therefore, discarding the second half of the experiment would not affect the associative learning processes taking place in the first half of the experiment. Second, we did not include six of the 30 participants that are in the publicly available data set provided by *Rutar et al., 2023* due to missing cases in the repeated-measures ANOVAs in phase 1 of the experiment. Finally, *Rutar et al., 2023* only tested for signed prediction errors within an early time window (see their Supplementary Materials) but did not investigate any later time windows as we do in the current experiment.

## Data set #2: letter-color 2AFC task

### Participants and informed consent

The final sample consisted of 47 participants aged 17–45 years ($M$ = 23.8, SD = 6.16, 34 women and 13 men). Fifty participants completed the experiment. Three participants had to be excluded due to technical error or human error on the part of the researcher. All participants gave written informed consent before participating and were compensated for participation.

### Tasks and procedure

The experiment consisted of two separate tasks, the odd-ball detection and 2AFC decision tasks, which corresponded to a learning and decision-making phase, respectively (*Figure 2A*). Participants

were not instructed about the decision-making task until the learning task was completed. A chin rest was used to keep the distance (58 cm) to the computer screen constant (1920 x 1080 pixels, 120 Hz). This setup resulted in visual stimuli spanning visual angles between 2.6° and 3.1°. We exposed participants to pairs of stimuli presented together in different frequency conditions during the learning phase of the experiment. We aimed to have low, medium, and high levels of stimulus-pair frequency to correspond to different amounts of exposure to associations between the individual letters and the colors. More exposure to consistent associations between letters and colors was expected to result in stronger expectations of the specific letter-color pairs as compared with less exposure. After this odd-ball task in which participants were exposed to the letter-color pairs in different frequency conditions, participants were asked to make a 2AFC decision based on the presented stimuli pairs in the decision-making phase. Participants completed a questionnaire at the end of the computer tasks (data not reported here). Participants were given self-paced breaks between each task. The entire session took about 2.5 hr to complete.

### Independent learning phase: odd-ball detection task

We hypothesized that statistical learning would take place within an odd-ball detection paradigm, during which participants had to monitor both the identity of a letter or number and the background color it was presented on (see *Figure 2A*). The stimuli used for the statistical-learning hypothesis were six letters ('A', 'D', 'I', 'O', 'R', and 'T', 100 pixels, Bookman Old Style font) and six shades of green as the background color in the shape of a square (120 x 120 pixels; see *Figure 2A* and *Appendix 3—figure 1* for hexadecimal codes). Shades of a single hue were chosen to help minimize verbal heuristic strategies for naming the six different colors. Three different frequency conditions were used (20%, 40%, and 80% of trials in which a specific letter was presented against a square background with a specific color), meaning that two letter-color pairs were in each frequency condition. For example, if the letter 'A' was assigned to the 80% frequency condition, then 'A' was shown with its associated shade of green as the background color in eight out of 10 trials in which 'A' was presented. On the remaining two out of 10 trials in which an 'A' was presented, the background shade of green was randomly drawn from all six shades of green. This sampling with replacement resulted in the letters being shown together with their associated color in actual frequency conditions of 33%, 50%, and 84%. Each letter was shown together with the other five unassociated colors on average in 13%, 10%, and 3% of trials (i.e. the noise around the letter-color pair signal) with respect to the 20%, 40%, and 80% frequency conditions, respectively. The six-letter-color pair combinations as well as their frequency condition were randomly assigned to each participant at the start of the experiment, meaning that participants received individual combinations of stimuli. For clarity, we will only refer to the actual frequency conditions in the letter-color 2AFC task (i.e. 33%, 50%, and 84%) from here on instead of the intended frequency conditions (20%, 40%, and 80%).

The odd-ball stimuli were numbers 1–9 (randomly drawn) and a non-green hue (one of four colors was chosen by random for each participant at the start of the experiment). Note that odd-balls could consist of (i) a number with a non-green background, (ii) a number with a green background, and (iii) a letter with a non-green background. Participants completed a short round of 10 practice trials of the odd-ball task, which they could repeat until they understood the odd-ball rule. During the practice round, an equal number of odd-ball and regular stimuli were presented. Participants were instructed to indicate whether the stimulus on each trial was an odd-ball or not with either a left or a right button press (button order was counterbalanced between participants). They could respond as soon as the stimulus appeared on screen. Participants were not informed about the frequency conditions of the letter-color pairs. The odd-ball task consisted of 660 trials in total (9% were odd-balls), during which participants had two self-paced breaks. The order of the trials was randomly presented to each participant. The odd-ball task took about 30 min to complete.

### Oddball-task trial structure

Each trial of the odd-ball task consisted of a stimulus period (0.75 s), a response period (<1.25 s), and an inter-trial interval that included feedback on accuracy (0.5–1 s jittered). A black fixation cross was presented in the center of the screen and changed to green, red, or blue in the inter-trial interval for correct, incorrect, or missed responses, respectively. Errors and missed trials were also accompanied

by a short auditory tone (3rd octave 'D' for 0.3 s). All stimuli were presented in the center of the screen against a gray background.

## Letter-color 2AFC task

After the odd-ball task, participants performed a 2AFC decision-making task on the occurrence of letter-color pairs that were shown during the preceding odd-ball task. Specifically, participants were instructed to indicate whether specific letter-color pairs occurred most often together in the preceding odd-ball task. They were instructed to guess if they were unsure. If a letter occurred most often together with a particular color in the preceding oddball task, then this was considered a 'match'. The 2AFC response options were 'match' or 'no match'. For example, if the letter 'A' was most often shown together with a specific shade of green as the background color during the preceding odd-ball task, then 'A' would *match* this shade of green. Match and no-match conditions were presented in a 1:1 ratio of the number of trials. Participants could respond as soon as the color was presented on screen. Participants were instructed to indicate whether the letter-color pairs matched or not with either a left or a right button press (button order was counterbalanced between participants). The letter-color visual decision task consisted of 250 trials in total, during which participants had three self-paced breaks. The order of the trials was randomly presented to each participant.

## Letter-color 2AFC task trial structure

Each trial of the letter-color visual decision task consisted of a new trial cue period (0.2 s), a letter-stimulus period (0.75 s), a short delay period (0.1 s), a response period during which a colored square appeared (<2.5 s), a longer delay period to give sufficient time for the pupil to return to baseline following a colored impulse (3.5–5.5 s, uniform distribution; see *Appendix 1—figure 1A, B*; *Mathot, 2018*), and an inter-trial interval that included feedback on accuracy (3.5–5.5 s, uniform distribution). A black fixation cross was presented in the center of the screen during the longer delay period preceding feedback as well as in the inter-trial interval. Trial-by-trial feedback was presented to the participants by means of two auditory tones (0.3 s) for errors (3rd octave 'D') and correct trials (4th octave 'B'). We verified that the tone-locked pupil response was not differentially affected by the two feedback tones irrespective of the task context (see *Appendix 3—figure 1C, D*). Participants were familiarized with the feedback tones before the decision task began. All stimuli were presented in the center of the screen against a gray background. The letter-color decision 2AFC task took about one hour to complete.

## Data acquisition and preprocessing

Pupil dilation of the left or right eye was continuously recorded during the letter-color visual decision task using an EyeLink eye-tracker (SR Research, Ottawa, Ontario, Canada) with a sampling rate of 1000 Hz. The eye-tracker was calibrated once at the start of the decision task. A drift correction was performed after participants took breaks and could move their heads before continuing. Eye blinks and saccades were detected using the standard EyeLink software algorithms (default settings). We aimed to keep the pre-processing pipelines between the two data sets as similar as possible. Using custom Python code, the following steps were applied to the entire pupil dilation time series: (i) linear interpolation around missing samples (0.15 s before and after each missing event), (ii) linear interpolation around blinks or saccade events based on spikes in the temporal derivative (0.15 s before and after each nuisance event), (iii) band-pass filtering (third-order Butterworth, 0.01–6 Hz), (iv) responses to nuisance events were removed using linear regression (nuisance responses were estimated using deconvolution; *Knapen et al., 2016*) and (v) the residuals of the nuisance regression were converted to percent signal change with respect to the temporal mean. The size of the interpolation window preceding nuisance events was based on previous literature (*Urai et al., 2017*; *Knapen et al., 2016*; *Winn et al., 2018*). After interpolation based on data markers and/or missing values, remaining blinks and saccades were estimated by testing the first derivative of the pupil dilation time series against a threshold rate of change. The threshold for identifying peaks in the temporal derivative is data-driven, partially based on past work (*Colizoli et al., 2018*; *de Gee et al., 2017*; *Rutar et al., 2023*). The output of each participant's pre-processing pipeline was checked visually. Once an appropriate threshold was established at the group level, it remained the same for all participants (minimum peak height of 10 units). Missed trials were excluded from all analyses (0.9%).

## Quantification of the feedback-locked pupil response

After pre-processing of the pupil data in both tasks, each trial time course was baseline corrected in percent signal change units. The baseline window was defined per trial as the mean pupil size during the 0.5 s before the feedback event (target onset or auditory feedback for the cue-target and letter-color 2AFC tasks, respectively; *Colizoli et al., 2018*; *de Gee et al., 2021*; *Rutar et al., 2023*). The same time windows were used for both tasks with respect to the feedback event. Feedback-locked pupil responses were defined by the mean pupil response within two time windows of interest: An early time window was defined to be 0.75–1.25 seconds after feedback onset to be centered around the peak of a transient event based on the canonical impulse response function of the pupil (*Colizoli et al., 2018*; *de Gee et al., 2021*; *de Gee et al., 2014*; *Hoeks and Levelt, 1993*). A late time window was defined to be 2.5–3 s after feedback onset and was determined by the shortest feedback interval within both decision tasks to make sure that the pupil response was uncontaminated by the subsequent trial: in the cue-target 2AFC task, a new trial started 3 s after the feedback onset. Several sanity checks related to the pre-processing pipeline were carried out on the feedback-locked pupil response (see Appendix 4). The raw and interpolated time courses are shown before the (blink and saccade) nuisance regression (*Appendix 4—figure 1*, top two rows, respectively). We present a conservative analysis in which only trials with more than half of original (i.e. non-interpolated) data are included in the analyses (*Appendix 4—figure 1*, third row). Finally, the nuisance predictor time courses (based on blink and saccade events) are shown for the same conditions as the feedback-locked pupil response (*Appendix 4—figure 1*, bottom row).

## Ideal learner models

Following previous literature (*O'Reilly et al., 2013*; *Mars et al., 2008*; *Poli et al., 2020*), we adapted an ideal learner model to each of the two datasets separately. To model the learning of the task by participants, we assumed that they acted as ideal observers who learn the probability of seeing each of the stimulus pair types as the experiment progressed across trials. For each decision-making task, trial-by-trial quantifications of KL divergence between the posterior and prior belief distributions were estimated using information theory to formally quantify information gain. In addition, surprise and entropy were computed and assumed to reflect the subjective probability and average uncertainty on each trial, respectively.

The ideal learner model for both data sets follows from the algorithms described by *Mars et al., 2008*; *Poli et al., 2020*. The ideal learner model represents a set of discrete events $x$, which can range from one to $K$. The events follow each other in a sequence with its length, $j$, equal to the number of trials in the task and denoted by $X^j = \left\{ x^1, \ldots, x^j \right\}$. We assume the participants learn the probabilities of $X^j$ occurring, and following each trial, they update their estimate of the probability that each event type will occur based on the previously observed events. The distribution of probabilities can be parametrized by the vector $P(x) = [p_1, \ldots, p_K]$, the elements of which sum to one and will be abbreviated by $P(x) = p$. Note that the probability of the $k$ th event occurring, $P(x = k)$, is denoted by $p_k$.

Following previous literature (*Mars et al., 2008*; *Poli et al., 2020*), we assumed a Bayesian paradigm of giving the observer prior knowledge represented by a prior distribution of beliefs. This takes the form of a Dirichlet distribution which indicates the belief in all parameters prior to any observations of events. A Dirichlet prior over $p$ is parameterized by the vector $\alpha = [\alpha_1, \ldots, \alpha_K]$ and denoted by $P(p|\alpha) = Dir(p; \alpha_k)$. All elements of $\alpha$ are positive and the magnitude of each element corresponds to the relative expectation of each element. When the ideal learner has no previous expectations about which event will occur before the first trial of the task begins, all events are considered equally likely to occur and all the elements of $\alpha$ would be set to one (i.e. a uniform prior distribution). For example, if $\alpha = [100, 1, 1, 1]$, then the ideal learner model assumes that the event represented by the first element in $\alpha$ is much more likely to occur than any other event.

After an event is observed, the estimated probabilities $p$ will be updated. The belief after $j$ trials, $X^j$, is given by the posterior distribution

$$P\left( p|X^j, \alpha \right) = Dir\left( p^j; n_k^j + \alpha_k \right) = D^j, \tag{1}$$

in which $n_k^j$ refers to the number of occurrences of event type $k$ until trial $j$, and the parameter, $\alpha_k$, determines prior expectations of event type $k$ occurring. The posterior distribution given in

**Equation 1** is updated after each new observation of an event and is abbreviated as $D^j$. For example, if $\alpha = \begin{bmatrix} 100, 1, 1, 1 \end{bmatrix}$ and the participant observes an event represented by the second element in $\alpha$ on the first trial, then the posterior distribution becomes $\begin{bmatrix} 100, 2, 1, 1 \end{bmatrix}$ for $j = 1$.

The probability of a certain event occurring at trial $j$ can be computed directly from (1) as

$$p\left(x^j = k | X^{j-1}, \alpha\right) = \frac{\left(n_k^{j-1} + \alpha_k\right)}{(j - 1 + K)} = \widetilde{p}_k^j \tag{2}$$

In words, **Equation 2** states that the probability that a certain event $k$ will occur on trial $j$ is denoted by $\widetilde{p}_k^j$ (note the tilde indicates a prediction), which is equal to the total number of times event $k$ occurred in previous trials $j - 1$ plus the value of $\alpha_k$, which is known to be fixed in the prior distribution, then divided by the total number of observations up to and including $j - 1$ plus the total number of possible event types $K$.

## Information-theoretic variables

Information theory (**Mars et al., 2008**; **Poli et al., 2020**; **Attneave, 1959**; **Shannon, 1948**; **Strange et al., 2005**) is used to estimate the information gain, surprise, and entropy at each trial. The subjective probability of each event occurring was quantified as surprise in terms of Shannon information, $I$

$$I\left(x^j = k\right) = -log_2 \widetilde{p}_k^j. \tag{3}$$

Note that $\widetilde{p}_k^j$ is given by **Equation 2** and represents the prediction of the probability of event $k$ occurring at trial $j$ based on what has been observed up to and including trial $j - 1$. The negative logarithm ensures that highly probable events are considered as less surprising.

The average uncertainty of the event at each trial was quantified as the entropy, $H$,

$$H\left(p^j\right) = -\sum_{k=1}^{k} p_k^j log_2 p_k^j. \tag{4}$$

Note that entropy was calculated including the observation of the event at trial $j$, denoted by $p_k^j$, not just up to trial $j - 1$, because the participants received the information related to trial $j$ at the time of feedback presentation.

Finally, the information gain at each trial was quantified as the KL divergence, $D_{KL}$,

$$D_{KL}^j\left(p^j | p^{j-1}\right) = \sum_{k=1}^{k} p_k^j log_2 \frac{p_k^j}{p_k^{j-1}}, \tag{5}$$

between the posterior and prior distributions at trial $j$. Note that this KL divergence is sometimes referred to as a 'Bayesian surprise' (**Itti and Baldi, 2005**; **Itti and Baldi, 2009**). Here, when referring to 'surprise', we are always referring to the Shannon information as given by **Equation 3**.

## Ideal learner model assumptions for data set #1: cue-target 2AFC task

In the cue-target 2AFC task, an event was defined as one of the four possible cue-target pairs determined by the two cues (square and diamond) and two target orientations (left and right). Therefore, in the cue-target 2AFC task, $K = 4$, and $x$ could take on values from 1 to 4. We assumed that the participants had no prior knowledge about the probabilities of events occurring at the start of the cue-target 2AFC task, and therefore, all events are assumed to be equally likely in the prior distribution. Since there are four possible events that can occur in the cue-target 2AFC task, then $P\left(x\right) = \begin{bmatrix} p_1, p_2, p_3, p_4 \end{bmatrix}$ and $\alpha = \begin{bmatrix} 1, 1, 1, 1 \end{bmatrix}$. Note that this resulted in a belief that all four cue-target pairs were likely to occur 25% of the time. We also note that while the ideal learner model for the cue-target 2AFC task used a uniform (flat) prior distribution for all participants, the model parameters were based on the participant-specific cue-target counterbalancing conditions and randomized trial order.

### Ideal learner model assumptions for data set #2: letter-color 2AFC task

For the letter-color 2AFC task, during which pupil dilation was recorded, there were six letters and six shades of green used as stimuli. An event was defined as one of the 36 possible letter-color pairs; therefore, $K = 36$, and $x$ could take on values from one to 36 and $P(x) = [p_1, \ldots, p_{36}]$. We assumed that the participants started the letter-color 2AFC task with prior beliefs about the probabilities of the 36 letter-color pairs based on their observations during the preceding odd-ball task. The prior distribution for the letter-color 2AFC ideal learner model was given by $\alpha = P_o(x) = [p_1, \ldots, p_{36}]$, where $P_o$ refers to the final probabilities of all 36 event types occurring at the end of the odd-ball task. Thus, the prior distributions used for the letter-color 2AFC task were estimated from the randomized letter-color pairs and randomized trial order in the preceding odd-ball task; this resulted in participant-specific prior distributions for the ideal learner model of the letter-color 2AFC task. The model parameters were likewise estimated from the (participant-specific) randomized trial order presented in the letter-color 2AFC task. Note that the probabilities of the odd-ball stimuli (i.e., an odd color or number) occurring were not estimated in the model. Unlike in the odd-ball task, the 36 letter-color pairs occurred at equal frequency in the letter-color 2AFC task; the occurrence of response options (match vs. no match) was also balanced. Learning about the letter-color probabilities could still occur, however, through the presentation of trial-by-trial feedback on accuracy.

### Software and statistical analysis

All data and code used are publicly available (see link in data availability statement). All tasks were presented with PsychoPy (cue-target 2AFC task, version 1.81; letter-color 2AFC, version 1.82). Custom software in Python (version 3.6) was used for the preprocessing and data analysis with the exception of the repeated-measures ANOVAs, t-tests, and linear mixed modeling analysis (see below). Note that the (*Rutar et al., 2023*) data set has been re-analyzed here and the above refers to the corresponding code for the current study. Python-specific packages and versions used are listed in the code repository.

Statistical inference on the evoked pupil responses was conducted with cluster-based permutation test from the MNE-Python package (*Gramfort et al., 2013*) unless otherwise stated. When the cluster-based permutation test could not be applied to a simple difference between two conditions, the evoked pupil responses were corrected with the false discovery rate instead. Repeated-measures ANOVAs and paired-samples t-tests were conducted using JASP version 0.16.3 (*JASP Team, 2022*). Wilcoxon signed-ranked t-tests were used when normality was deviated, and effect size is indicated by the matched rank biserial correlation ($r_{rb}$). For ANOVA results, Greenhouse-Geisser statistics are reported for violations of the assumption of sphericity and generalized eta squared ($\eta^2_G$) is reported as the effect size. In repeated-measures designs, $\eta^2_G$ is useful as it can be more easily compared with between-subject designs (*Lakens, 2013*; *Olejnik and Algina, 2003*).

### Comparing the information-theoretic variables to the feedback-locked pupil response

We sought to examine the overall relationship each of the information-theoretic variables has with the post-feedback pupil response. Furthermore, it was expected that these explanatory variables would be correlated with one another. For this reason, we pursued two complementary approaches to the model fitting (*Morrissey and Ruxton, 2018*): a 'simple' correlation analysis and a linear mixed model comparison. Both the correlation analysis and linear mixed model comparison were done for all trials and then separately for error and correct trials.

First, in the correlation analysis, for each decision task and per participant, we computed the trial-by-trial correlations between the feedback-locked pupil response at each time point in the time course (i.e. in the interval 0–3 s) and the surprise, entropy, and information gain in bits. Correlation coefficients were normalized using the Fisher transform (*Fisher, 1915*) then averaged across participants. Statistical significance of the coefficients at the group level was assessed using non-parametric cluster-based permutation tests (MNE-Python package).

Second, in a complementary linear mixed model analysis, we compared two linear models to evaluate which combination of predictor variables provided the best fit to the feedback-locked pupil response as the dependent variable. Each model fit included the data for all participants and was done independently for each decision task and time window of interest (early and late). Model 1

consisted of the three model parameters, surprise, entropy, and information gain. Model 2 was identical to Model 1 except that an interaction term between entropy and information gain was additionally included. The pre-feedback baseline pupil dilation and reaction times were included as additional predictor (nuisance) variables in both models for the following reasons: The pre-feedback baseline pupil dilation was expected to improve the overall model fits to the feedback-locked pupil response, as event-related pupil responses are partially determined by the size of the proceeding baseline pupil dilation (*de Gee et al., 2014*). Both simple motor actions as well as statistical confidence/uncertainty about decisions have been shown to scale with the response-locked pupil signal (*Colizoli et al., 2018*; *Urai et al., 2017*; *Hoeks and Levelt, 1993*; *Hupé et al., 2009*). Therefore, reaction time differences may have contributed to the variation in the feedback-locked pupil response; although we note that both tasks included relatively long inter-stimulus intervals (at least 3 s) following the motor responses on each trial before feedback was presented to allow the pupil response sufficient time to return to baseline. Both models included a constant term and 'subject' as a random effect; all other effects were modeled as fixed effects. Data were modeled with Bayesian regression modeling (*brms*) using the *brm* function of the *brms* package in the software package R (R Studio Version 2024.12.1+563) with 10,000 iterations, 5000 warm-ups, and four chains (*Bürkner, 2017*). Other settings used were the default parameters unless otherwise stated. We confirmed that all chains converged to the same posterior distribution (all Rhats = 1). Model comparison was performed using Pareto smoothed importance sampling and leave-one-out cross-validation in the *looic* toolbox in R (*Vehtari et al., 2017*). The *loo* function automatically designates the best-performing model with the highest expected log predictive density (ELPD) as the 'standard' model. To assess relative model performance between the models, the ratio between the difference in ELPD and the difference in the corresponding standard error was computed: ratios greater than two were considered to be statistically meaningful (*Kelter, 2021*; *Merkle and Rosseel, 2018*). The linear models that were compared to each other are given below in their equation format, where $I$ represents Shannon surprise, $H$ represents entropy, and $D_{KL}$ represents information gain:

> Model 1: pupil ~ $I + H + D_{KL}$ + pre-feedback baseline pupil + reaction time + (1|subject)
> Model 2: pupil ~ $I + H*D_{KL}$ + pre-feedback baseline pupil + reaction time + (1|subject)

## Acknowledgements

We would like to thank and acknowledge Yibrán Amador Pacheco Sáez, Filip Novický, Francesco Poli, Luke Miller, Lieke van Lieshout, Anna Umurzakova, and the Karl Friston Lab for helpful insight and discussions at various stages of the research presented here. We would also like to thank the reviewers and editors who helped improve the manuscript with their constructive feedback. This research was funded by the Donders Institute for Brain, Cognition and Behaviour, Radboud University Nijmegen, The Netherlands.

## Additional information

### Funding

No external funding was received for this work.

### Author contributions

Olympia Colizoli, Conceptualization, Data curation, Software, Formal analysis, Investigation, Visualization, Methodology, Writing – original draft, Project administration, Writing – review and editing; Tessa M van Leeuwen, Conceptualization, Investigation, Methodology, Writing – review and editing; Danaja Rutar, Data curation, Investigation, Methodology, Writing – review and editing; Harold Bekkering, Conceptualization, Supervision, Writing – review and editing

### Author ORCIDs

Olympia Colizoli https://orcid.org/0000-0001-5288-2437
Tessa M van Leeuwen https://orcid.org/0000-0001-7810-6348
Danaja Rutar https://orcid.org/0000-0002-6798-2796

Harold Bekkering  https://orcid.org/0000-0001-8957-4817

### Ethics

All experiments were conducted on human participants and were approved by the Ethical Committee of the Faculty of Social Sciences at the Radboud University, the Netherlands (data set #1: ECSW2016-0905-396; data set #2: ECSW-2018-115).

Reviewer #1 (Public review): https://doi.org/10.7554/eLife.105287.3.sa1
Reviewer #2 (Public review): https://doi.org/10.7554/eLife.105287.3.sa2
Reviewer #3 (Public review): https://doi.org/10.7554/eLife.105287.3.sa3
Author response https://doi.org/10.7554/eLife.105287.3.sa4

## Additional files

### Supplementary files
MDAR checklist

### Data availability

All behavioral and physiological data for data set #2, along with the the code for the analyses of both data sets, are openly accessible at the Radboud University Data Repository: https://doi.org/10.34973/dt44-x857. Data set #1 was previously published and the article is available at: https://doi.org/10.1371/journal.pone.0270619.

The following previously published dataset was used:

| Author(s) | Year | Dataset title | Dataset URL | Database and Identifier |
|---|---|---|---|---|
| Rutar D, Colizoli O, Selen LPJ, Spieß L, Kwisthout JHP | 2023 | Differentiating between Bayesian parameter learning and structure learning based on behavioural and pupil measures | https://doi.org/10.34973/t41p-hx94 | Radboud University Data Repository, 10.34973/t41p-hx94 |

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

# Appendix 1

## Individual differences analysis between accuracy and pupil responses

### Data set #1: cue-target 2AFC task

In data set #1, it was previously reported that pupil responses decreased for the high-frequency (expected) trials as compared with low-frequency (unexpected) over the course of the experiment as a result of learning the cue-target contingencies (*Rutar et al., 2023*; see their Supplementary Figure 2). In the current analysis, we explored whether a similar relationship between the feedback-locked pupil response and behavioral accuracy was also evident at the level of individual differences across the participant sample. We similarly expected that the feedback-locked pupil response and accuracy would 'mirror' one another in such a way that those individuals who showed a larger difference between the 80% and 20% frequency conditions in accuracy would also show the larger (absolute) difference between frequency conditions in the feedback-locked pupil response. Within the early time window, a significant negative Spearman correlation was obtained indicating that individuals who showed larger differences between the 80% as compared with the 20% frequency conditions in accuracy also showed smaller differences between frequency conditions in the feedback-locked pupil response (*Appendix 1—figure 1*, top left). The negative direction of the correlation can be explained because the pupil responses are larger on average for the 20% frequency condition as compared with the 80% frequency condition (*Figure 1—figure supplement 1C*). A trend toward the same negative scaling could be seen in the late time window (*Appendix 1—figure 1*, top right).

### Data set #2: letter-color 2AFC task

We explored whether this relationship between the feedback-locked pupil response and behavioral accuracy was also evident at the level of individual differences during the letter-color 2AFC task. However, in contrast to the cue-target 2AFC task, we did not find any evidence that the frequency effect in task accuracy scaled with the frequency effect in the feedback-locked pupil response in either the early or late time window (*Appendix 1—figure 1*, bottom row).

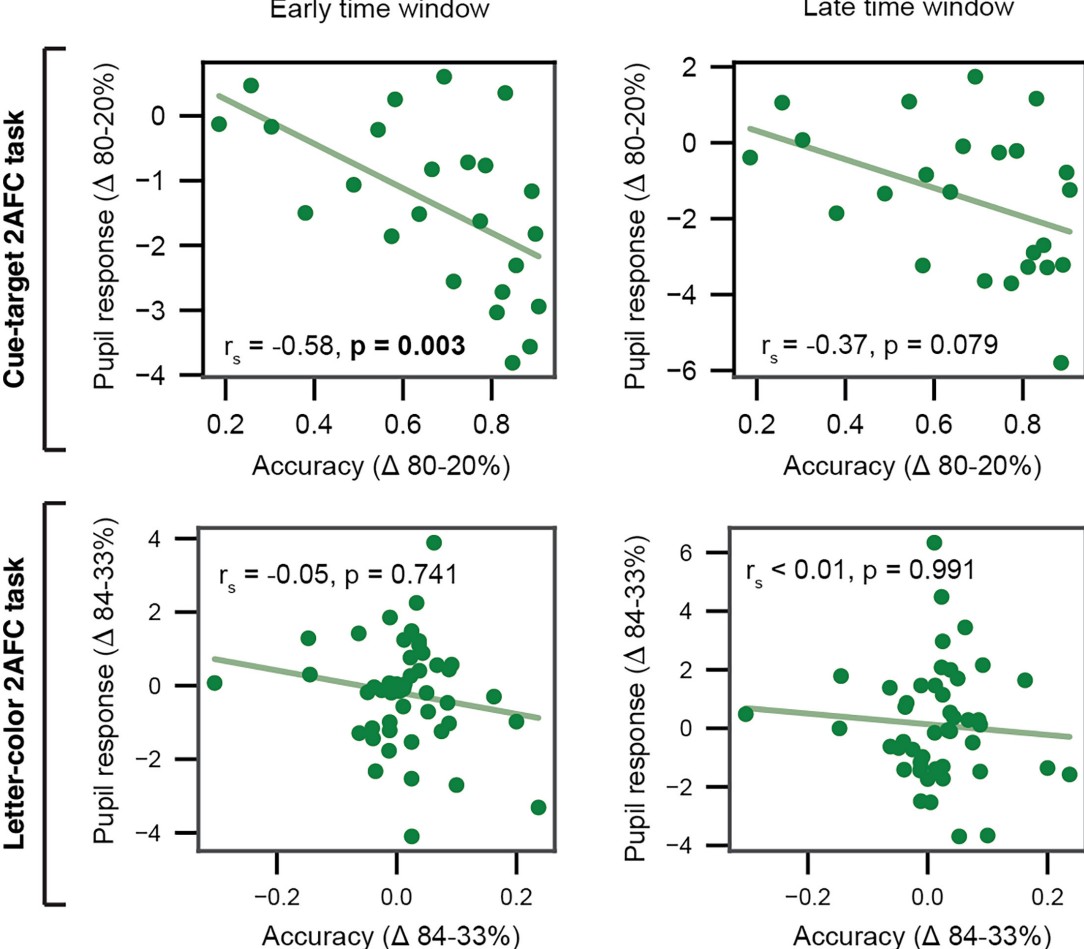

**Appendix 1—figure 1.** Individual differences analysis between accuracy and pupil responses. *Top row*, cue-target 2AFC task. *Bottom row*, letter-color 2AFC task. *Left column*, early time window. Right column, late time window. The average feedback-locked pupil response frequency difference (80–20% and 84–33% frequency conditions for the cue-target and letter-color 2AFC tasks, respectively) is plotted against the frequency difference in accuracy. Data points, individual participants.

The online version of this article includes the following source data for appendix 1—figure 1:

**Appendix 1—figure 1—source data 1.** Processed data for the individual differences analysis between accuracy and pupil responses (both tasks).

## Appendix 2

### Linear mixed model comparisons

The *loo* function (in the *looic* toolbox in R) automatically designates the best-performing model with the highest expected log predictive density (EPLD) as the 'standard' model (see the first row of each model comparison). To assess relative model performance between the models, the ratio between the difference in ELPD and the difference in the corresponding standard error was computed: ratios greater than two were considered to be statistically meaningful (*Vehtari et al., 2017*; *Merkle and Rosseel, 2018*). Explanation of abbreviations: *ELPD loo,* expected log predictive density. This is the sum of the pointwise log-likelihoods for each observation, estimated using leave-one-out (LOO) cross-validation. Higher is better, indicating better out-of-sample predictive performance; *p loo*, effective number of parameters (estimated by LOO). This is not the literal number of parameters, but a complexity measure. Higher values suggest more flexible (or more overfit) models; *looic,* LOO information criterion that is on the deviance scale. Lower *looic* values are preferred; *SE*, Standard error.

**Appendix 2—table 1.** Linear mixed model comparisons for the cue-target 2AFC task.

| | **Cue-target 2AFC task** | | | | | | |
|---|---|---|---|---|---|---|---|
| | Early time window | | | | | | |
| | Model | ELPD loo (SE) | p loo (SE) | looic (SE) | △ELPD | △SE | \|△ELPD\|/△SE |
| | 1 | −13596.9 (77) | 30.3 (1.3) | 27195.5 (154) | 0.0 | 0.0 | 0.0 |
| All trials | 2 | −13598.5 (77) | 31.0 (1.3) | 27197.1 (154) | −0.8 | 0.4 | 2.0 |
| | 1 | −9499.1 (68.7) | 29.7 (1.7) | 18998.3 (137.3) | 0.0 | 0.0 | 0.0 |
| Correct trials | 2 | −9499.0 (68.7) | 30.0 (1.7) | 18998.1 (137.3) | −0.1 | 1 | 0.1 |
| | 2 | −4069.5 (35.1) | 29.8 (1.7) | 8139.1 (70.3) | 0.0 | 0.0 | 0.0 |
| Error trials | 1 | −4070.4 (35.2) | 28.8 (1.7) | 8140.8 (70.4) | −0.9 | 1.9 | 0.5 |
| | Late time window | | | | | | |
| | Model | ELPD loo (SE) | p loo (SE) | looic (SE) | △ELPD | △SE | \|△ELPD\|/△SE |
| | 1 | −14983.7 (67.7) | 29.7 (1.1) | 29967.4 (135.5) | 0.0 | 0.0 | 0.0 |
| All trials | 2 | −14984.7 (67.7) | 29.7 (1.1) | 29969.4 (135.5) | −1.0 | 0.3 | 3.3 |
| | 1 | −10575.0 (57.1) | 27.7 (1.2) | 21150.1 (114.3) | 0.0 | 0.0 | 0.0 |
| Correct trials | 2 | −10575.8 (57.2) | 29.0 (1.3) | 21151.5 (114.3) | −0.7 | 1.2 | 0.6 |
| | 2 | −4363.2 (33.3) | 27.1 (1.4) | 8726.3 (66.6) | 0.0 | 0.0 | 0.0 |
| Error trials | 1 | −4364.0 (33.4) | 26.7 (1.4) | 8728.0 (66.7) | −0.8 | 1.3 | 0.6 |

The online version of this article includes the following source data for appendix 2—table 1:

**Appendix 2—table 1—source data 1.** Processed data input into the linear mixed modeling analysis for the cue-target 2AFC task.

**Appendix 2—table 2.** Linear mixed model comparisons for the letter-color 2AFC task.

| | **Letter-color 2AFC task** | | | | | | |
|---|---|---|---|---|---|---|---|
| | Early time window | | | | | | |
| | Model | ELPD loo (SE) | p loo (SE) | looic (SE) | △ELPD | △SE | \|△ELPD\|/△SE |
| | 1 | −39805.3 (104.8) | 48.7 (1.1) | 79610.7 (209.6) | 0.0 | 0.0 | 0.0 |
| All trials | 2 | −39805.7 (104.8) | 49.3 (1.1) | 79611.4 (209.6) | −0.4 | 0.7 | 0.6 |
| | 1 | −32665.1 (95.8) | 45.1 (1.2) | 65330.2 (191.5) | 0.0 | 0.0 | 0.0 |
| Correct trials | 2 | −32666.0 (95.8) | 45.9 (1.2) | 65331.9 (191.5) | −0.9 | 0.2 | 4.5 |

*Appendix 2—table 2 Continued on next page*

*Appendix 2—table 2 Continued*

| | | **Letter-color 2AFC task** | | | | | |
|---|---|---|---|---|---|---|---|
| | 2 | −7024.7 (41.2) | 47.4 (2.4) | 14049.5 (82.4) | 0.0 | 0.0 | 0.0 |
| Error trials | 1 | −7025.6 (41.3) | 46.4 (2.4) | 14051.3 (82.6) | −0.9 | 1.8 | 0.5 |
| | | Late time window | | | | | |
| | Model | ELPD loo (SE) | p loo (SE) | looic (SE) | ΔELPD | ΔSE | \|ΔELPD\|/ΔSE |
| | 1 | −43351.0 (110.0) | 43.7 (1.0) | 86701.9 (220.0) | 0.0 | 0.0 | 0.0 |
| All trials | 2 | −43351.5 (110.0) | 44.4 (1.0) | 86703.0 (220.1) | −0.5 | 0.7 | 0.7 |
| | 1 | −35663.4 (97.0) | 40.9 (1.0) | 71326.8 (194.0) | 0.0 | 0.0 | 0.0 |
| Correct trials | 2 | −35664.2 (97.0) | 41.8 (1.0) | 71328.4 (194.0) | −0.8 | 0.5 | 1.6 |
| | 1 | −7628.3 (51.2) | 41.2 (2.8) | 15256.6 (102.5) | 0.0 | 0.0 | 0.0 |
| Error trials | 2 | −7628.7 (51.3) | 41.9 (2.8) | 15257.4 (102.6) | −0.4 | 0.7 | 0.6 |

The online version of this article includes the following source data for appendix 2—table 2:

**Appendix 2—table 2—source data 1.** Processed data input into the linear mixed modeling analysis for the letter-color 2AFC task.

## Appendix 3

### Control tasks for data set #2

Different colors and tones could influence the pupil response due to inherent properties of the stimuli, and thereby confound true feedback-related signals. Therefore, complementary to the main analysis, we administered two control tasks in one independent sample of participants to directly assess whether confounding effects on the pupil's response to the colors and tones presented in the letter-color 2AFC task should be expected.

### Participants and informed consent

An independent sample of fifteen participants ($M$ = 24.27 y, SD = 5.79, gender data was not acquired) completed both control tasks in a single session that lasted 30 min in total. All stimuli and equipment were identical to the setup reported for the letter-color 2AFC task. All participants gave written informed consent before participating and were financially compensated for participation with the standard amount.

### 3.1 The pupil's response to colored squares returns to baseline before feedback onset

#### Procedure and Methods

Color was a stimulus dimension in the letter-color 2AFC task, and colors are known to result in impulse responses driven by the light reflex of the pupil (*Barbur et al., 1992*). This light reflex is uninteresting in the context of the current experiment and furthermore could potentially affect the feedback-related pupil response of interest differently for different colors. Participants were presented with the six colored squares used in the letter-color 2AFC tasks and instructed to react with a button press as soon as the color appeared. In the main letter-color 2AFC task, the colored square was replaced with a fixation cross upon the participant's response, thus stimulus duration varied across trials and participants. To account for part of this variation in stimulus duration, each of the 15 control participants was yoked to one participant in the letter-color 2AFC task. In the control task for colors, the stimulus duration of the colored squares was equal to the mean reaction time (across all trials) of the yoked counterpart of each participant (group average: $M$ = 0.70 s, SD = 0.15). On each trial, a fixation cross was presented for 0.5 s, before the colored square was presented for a variable duration (see above), followed by the fixation cross again for the inter-trial interval (3.5–5.5 s; uniform distribution). Each color was presented 20 times for a total of 120 trials while pupil dilation was simultaneously recorded. Behavioral data on the control task were not analyzed.

#### Results

From the results, on average, the colors resulted in a constriction of the pupil dilation (*Appendix 3—figure 1A*). Crucially, when inspecting each of the six colors individually, all responses returned to baseline well before the delay period (3.5–5.5 s; uniform distribution) was terminated and the auditory feedback was presented (*Appendix 3—figure 1B*).

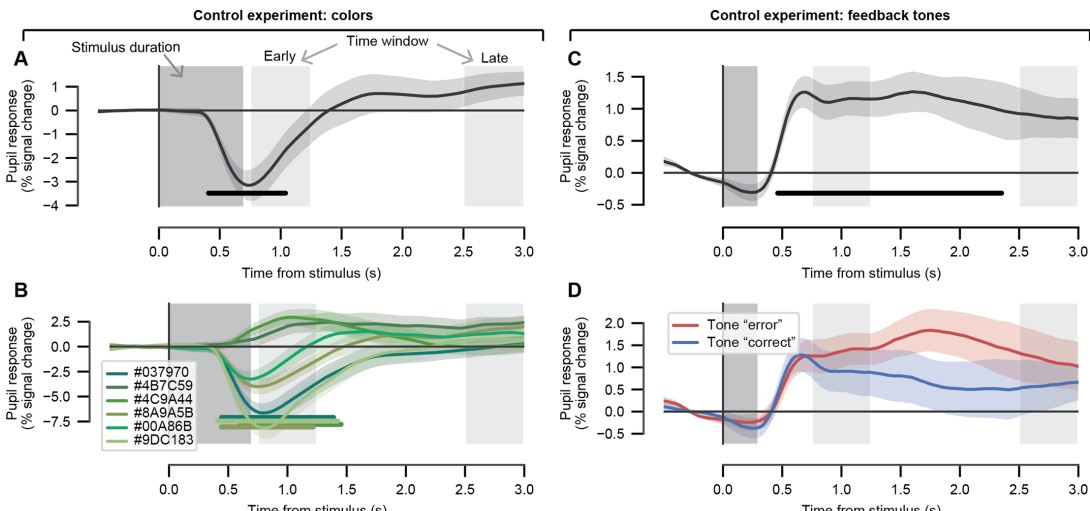

**Appendix 3—figure 1.** Control tasks for data set #2: letter-color 2AFC task. Left column, results from the control task for colors. Right column, results from the control task for feedback tones. (A) Mean tone-locked pupil response across all trials. (B) Feedback-locked pupil response time course plotted as a function of color used in the main letter-color 2AFC task (hexadecimal codes are given in the legend). (C) As A, for the control task for feedback tones. (D) Pupil response time courses plotted as a function of feedback tone used for error and correct trials in the main letter-color 2AFC task. All panels, dark gray boxes indicate the duration of the stimuli; 0.7 s, for the colors (group average); 0.3 s, auditory stimulus for the tones. Light gray boxes indicate time windows of interest; early time window, [0.75, 1.25]; late time window, [2.5, 3.0]. Shading represents the standard error of the mean across participants (N = 15). The black and green horizontal bars indicate a significant effect of interest (cluster-corrected, permutation based).

The online version of this article includes the following source data for appendix 3—figure 1:

**Appendix 3—figure 1—source data 1.** Processed pupil data used to generate the figures for the analysis of the control tasks related to the letter-color 2AFC task.

In sum, the control task for colors showed that the pupil's impulse response to the six different colors used in the letter-color 2AFC task would not have affected the upcoming feedback stimulus.

## 3.2 Mean pupil response is similar for the two 'feedback' tones

### Procedure and methods

The mapping of feedback tones to indicate accuracy on each trial was as follows: a high tone (4[th] octave "B") for correct trials and a low tone (3[rd] octave "D") for error trials. Thus, these tones were not counterbalanced across participants and could potentially confound a main effect of accuracy obtained in the pupil response (although we note that it could not account for any interaction obtained between accuracy and frequency conditions). Participants were presented with the two tones used as trial-wise feedback on accuracy in the letter-color 2AFC task and instructed to maintain their gaze on the fixation cross in the center of the screen for the duration of the task (no responses were required from the participants). On each trial, a fixation cross was presented for 0.5 s, before the tone was presented for 0.3 s, followed by the fixation cross again for the inter-trial interval (3.5 – 5.5 s; uniform distribution). Each tone was presented 50 times for a total of 100 trials while pupil dilation was simultaneously recorded.

### Results

Results showed that the auditory tone dilated pupils on average (*Appendix 3—figure 1C*). Crucially, however, the two tones did not differ from one another in either of the time windows of interest (*Appendix 3—figure 1D*); no significant time points after feedback onset were obtained either before or after correcting for multiple comparisons using cluster-based permutation methods; see Section 2.5.

In sum, the control task for feedback tones showed that the pupil responded similarly to the two different tones used in the letter-color 2AFC task. Thus, the different tone stimuli used to indicate error or correct trials would not have accounted for any differences obtained in the pupil's response on error as compared with correct trials.

# Appendix 4

## Sanity checks on pupil preprocessing

For visual comparison to the main results, the feedback-locked pupil response across the experimental conditions of interest is shown related to different pre-processing stages. These figures can be compared directly to *Figures 1D and 2D*, for the two tasks, respectively. The raw and interpolated time courses are shown before the (blink and saccade) nuisance regression. Both the raw and interpolated data have been band-pass filtered as was done in the original pre-processing pipeline and converted to percent signal change. The ratio of interpolated-to-original data (across the entire trial) varied greatly between participants and between trials: cue-target 2AFC task, M = 0.262, SD = 0.242, range = [0,1]; letter-color 2AFC task, M = 0.194, SD = 0.199, range = [0,1]; Here, we present a conservative analysis in which only trials with more than half (threshold = 60%) of original data are included in the analyses. Crucially, we still observe the same pattern of effects as when all data are considered across both tasks (compare the second to last row in *Appendix 4—figure 1* to *Figures 1D and 2D*). Note that for the cue-target 2AFC task, one participant did not have trials remaining in the four conditions of interest defined by the two-way interaction (frequency vs. accuracy) and was therefore removed for this post-hoc supplementary analysis. Finally, the nuisance predictor time courses (based on blink and saccade events) are shown for the same conditions as the feedback-locked pupil response. Model fits ($R^2$) for the nuisance regression were generally low: cue-target 2AFC task, M = 0.03, SD = 0.02, range = [0.00, 0.07]; letter-color 2AFC task, M = 0.08, SD = 0.04, range = [0.02, 0.16].

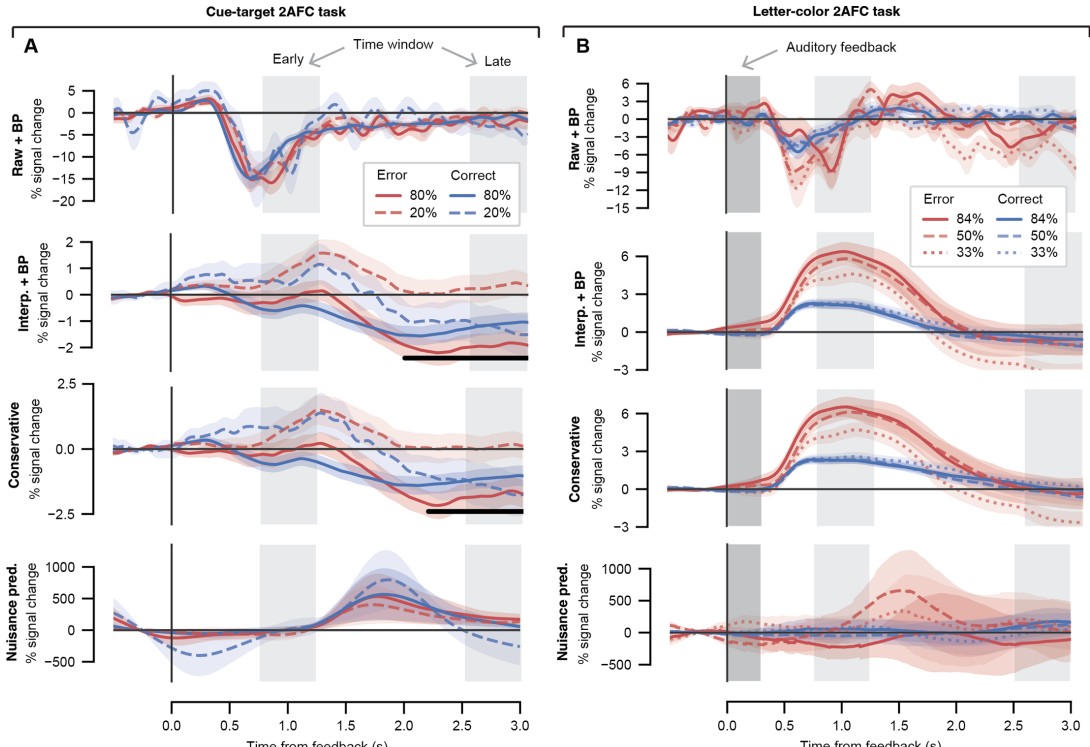

**Appendix 4—figure 1.** Sanity checks on pupil pre-processing for (**A**) the cue-target 2AFC task and (**B**) the letter-color 2AFC task. All plots, feedback-locked pupil response time course plotted as a function of cue-target frequency and accuracy for different pre-processing stages. Shading represents the standard error of the mean across participants (cue-target 2AFC task: N = 24; letter-color 2AFC task: N = 47). Light gray boxes, time windows of interest; early time window, [0.75, 1.25]; late time window, [2.5, 3.0]. The black horizontal bar indicates a significant interaction term (cluster-corrected, permutation test). Top row, the raw and band-pass filtered pupil signal before interpolation. Second row, the interpolated and band-pass filtered pupil signal but without the nuisance regression. Third row, the fully pre-processed pupil (as in the main results) for the conservative analysis in which only trials containing at least 60% of original (non-interpolated) data were included. Bottom row, the nuisance predictors based on blink and saccade events estimated by deconvolution.

The online version of this article includes the following source data for appendix 4—figure 1:

**Appendix 4—figure 1—source data 1.** Processed pupil data used to generate the appendix figure showing the sanity checks of pupil pre-processing figure for both tasks.

