## [Editor Report · eLife Assessment]

This **valuable** study investigates the relationship between pupil dilation and information gain during associative learning, using two different tasks. A key strength of this study is its exploration of pupil dilation beyond the immediate response period, extending analysis to later time windows after feedback, and it provides **convincing** evidence that pupillary response to information gain may be context-dependent during associative learning. The interpretation remains limited by task heterogeneity and unresolved contextual factors influencing pupil dynamics, but a range of interesting ideas are discussed.

---

## [Referee Report · Reviewer #1 (Public review)]

Summary:

This study examines whether changes in pupil size index prediction-error-related updating during associative learning, formalised as information gain via Kullback-Leibler (KL) divergence. Across two independent tasks, pupil responses scaled with KL divergence shortly after feedback, with the timing and direction of the response varying by task. Overall, the work supports the view that pupil size reflects information-theoretic processes in a context-dependent manner.

Strengths:

This study provides a novel and convincing contribution by linking pupil dilation to information-theoretic measures, such as KL divergence, supporting Zénon's hypothesis that pupil responses reflect information gain during learning. The robust methodology, including two independent datasets with distinct task structures, enhances the reliability and generalisability of the findings. By carefully analysing early and late time windows, the authors capture the timing and direction of prediction-error-related responses, offering new insights into the temporal dynamics of model updating. The use of an ideal-learner framework to quantify prediction errors, surprise, and uncertainty provides a principled account of the computational processes underlying pupil responses. The work also highlights the critical role of task context in shaping the direction and magnitude of these effects, revealing the adaptability of predictive processing mechanisms. Importantly, the conclusions are supported by rigorous control analyses and preprocessing sanity checks, as well as convergent results from frequentist and Bayesian linear mixed-effects modelling approaches.

Weaknesses:

Some aspects of directionality remain context-dependent, and on current evidence cannot be attributed specifically to whether average uncertainty increases or decreases across trials. Differences between the two tasks (e.g., sensory modality and learning regime) limit direct comparisons of effect direction and make mechanistic attribution cautious. In addition, subjective factors such as confidence were not measured and could influence both prediction-error signals and pupil responses. Importantly, the authors explicitly acknowledge these limitations, and the manuscript clearly frames them as areas for future work rather than settled conclusions.

---

## [Referee Report · Reviewer #2 (Public review)]

Summary:

The authors investigate whether pupil dilation reflects information gain during associative learning, formalised as Kullback-Leibler divergence within an ideal observer framework. They examine pupil responses in a late time window after feedback and compare these to information-theoretic estimates (information gain, surprise, and entropy) derived from two different tasks with contrasting uncertainty dynamics.

Strength:

The exploration of task evoked pupil dynamics beyond the immediate response/feedback period and then associating them with model estimates was interesting and inspiring. This offered a new perspective on the relationship between pupil dilation and information processing.

Weakness:

However, the interpretability of the findings remains constrained by the fundamental differences between the two tasks (stimulus modality, feedback type, and learning structure), which confound the claimed context-dependent effects. The later time-window pupil effects, although intriguing, are small in magnitude and may reflect residual noise or task-specific arousal fluctuations rather than distinct information-processing signals. Thus, while the study offers valuable methodological insight and contributes to ongoing debates about the role of the pupil in cognitive inference, its conclusions about the functional significance of late pupil responses should be treated with caution.

---

## [Referee Report · Reviewer #3 (Public review)]

Summary:

Thank you for inviting me to review this manuscript entitled "Pupil dilation offers a time-window on prediction error" by Colizoli and colleagues. The study examines prediction errors, information gain (Kullback-Leibler [KL] divergence), and uncertainty (entropy) from an information-theory perspective using two experimental tasks and pupillometry. The authors aim to test a theoretical proposal by Zénon (2019) that the pupil response reflects information gain (KL divergence). The conclusion of this work is that (post-feedback) pupil dilation in response to information gain is context dependent.

Strengths:

Use of an established Bayesian model to compute KL divergence and entropy.

Pupillometry data preprocessing and multiple robustness checks.

Weaknesses:

Operationalization of prediction errors based on frequency, accuracy, and their interaction:

The authors rely on a more model-agnostic definition of the prediction error in terms of stimulus frequency ("unsigned prediction error"), accuracy, and their interaction ("signed prediction error"). While I see the point, I would argue that this approach provides a simple approximation of the prediction error, but that a model-based approach would be more appropriate.

Model validation:

My impression is that the ideal learner model should work well in this case. However, the authors don't directly compare model behavior to participant behavior ("posterior predictive checks") to validate the model. Therefore, it is currently unclear if the model-derived terms like KL divergence and entropy provide reasonable estimates for the participant data.

Lack of a clear conclusion:

The authors conclude that this study shows for the first time that (post-feedback) pupil dilation in response to information gain is context dependent. However, the study does not offer a unifying explanation for such context dependence. The discussion is quite detailed with respect to task-specific effects, but fails to provide an overarching perspective on the context-dependent nature of pupil signatures of information gain. This seems to be partly due to the strong differences between the experimental tasks.

---

## [Author Response]

The following is the authors’ response to the current reviews.

**Public Reviews:**

**Reviewer #1 (Public review):**
Summary:This study examines whether changes in pupil size index prediction-error-related updating during associative learning, formalised as information gain via Kullback-Leibler (KL) divergence. Across two independent tasks, pupil responses scaled with KL divergence shortly after feedback, with the timing and direction of the response varying by task. Overall, the work supports the view that pupil size reflects information-theoretic processes in a context-dependent manner.Strengths:This study provides a novel and convincing contribution by linking pupil dilation to informationtheoretic measures, such as KL divergence, supporting Zénon's hypothesis that pupil responses reflect information gain during learning. The robust methodology, including two independent datasets with distinct task structures, enhances the reliability and generalisability of the findings. By carefully analysing early and late time windows, the authors capture the timing and direction of prediction-error-related responses, oPering new insights into the temporal dynamics of model updating. The use of an ideal-learner framework to quantify prediction errors, surprise, and uncertainty provides a principled account of the computational processes underlying pupil responses. The work also highlights the critical role of task context in shaping the direction and magnitude of these ePects, revealing the adaptability of predictive processing mechanisms. Importantly, the conclusions are supported by rigorous control analyses and preprocessing sanity checks, as well as convergent results from frequentist and Bayesian linear mixed-ePects modelling approaches.Weaknesses:Some aspects of directionality remain context-dependent, and on current evidence cannot be attributed specifically to whether average uncertainty increases or decreases across trials. DiPerences between the two tasks (e.g., sensory modality and learning regime) limit direct comparisons of ePect direction and make mechanistic attribution cautious. In addition, subjective factors such as confidence were not measured and could influence both predictionerror signals and pupil responses. Importantly, the authors explicitly acknowledge these limitations, and the manuscript clearly frames them as areas for future work rather than settled conclusions.
**Reviewer #2 (Public review):**
Summary:The authors investigate whether pupil dilation reflects information gain during associative learning, formalised as Kullback-Leibler divergence within an ideal observer framework. They examine pupil responses in a late time window after feedback and compare these to informationtheoretic estimates (information gain, surprise, and entropy) derived from two diPerent tasks with contrasting uncertainty dynamics.Strength:The exploration of task evoked pupil dynamics beyond the immediate response/feedback period and then associating them with model estimates was interesting and inspiring. This oPered a new perspective on the relationship between pupil dilation and information processing.Weakness:However, the interpretability of the findings remains constrained by the fundamental diPerences between the two tasks (stimulus modality, feedback type, and learning structure), which confound the claimed context-dependent ePects. The later time-window pupil ePects, although intriguing, are small in magnitude and may reflect residual noise or task-specific arousal fluctuations rather than distinct information-processing signals. Thus, while the study oPers valuable methodological insight and contributes to ongoing debates about the role of the pupil in cognitive inference, its conclusions about the functional significance of late pupil responses should be treated with caution.
**Reviewer #3 (Public review):**
Summary:Thank you for inviting me to review this manuscript entitled "Pupil dilation oPers a time-window on prediction error" by Colizoli and colleagues. The study examines prediction errors, information gain (Kullback-Leibler [KL] divergence), and uncertainty (entropy) from an information-theory perspective using two experimental tasks and pupillometry. The authors aim to test a theoretical proposal by Zénon (2019) that the pupil response reflects information gain (KL divergence). The conclusion of this work is that (post-feedback) pupil dilation in response to information gain is context dependent.Strengths:Use of an established Bayesian model to compute KL divergence and entropy.Pupillometry data preprocessing and multiple robustness checks.Weaknesses:Operationalization of prediction errors based on frequency, accuracy, and their interaction:The authors rely on a more model-agnostic definition of the prediction error in terms of stimulus frequency ("unsigned prediction error"), accuracy, and their interaction ("signed prediction error"). While I see the point, I would argue that this approach provides a simple approximation of the prediction error, but that a model-based approach would be more appropriate.Model validation:My impression is that the ideal learner model should work well in this case. However, the authors don't directly compare model behavior to participant behavior ("posterior predictive checks") to validate the model. Therefore, it is currently unclear if the model-derived terms like KL divergence and entropy provide reasonable estimates for the participant data.Lack of a clear conclusion:The authors conclude that this study shows for the first time that (post-feedback) pupil dilation in response to information gain is context dependent. However, the study does not oPer a unifying explanation for such context dependence. The discussion is quite detailed with respect to taskspecific ePects, but fails to provide an overarching perspective on the context-dependent nature of pupil signatures of information gain. This seems to be partly due to the strong diPerences between the experimental tasks.
**Recommendations for the authors:**

**Reviewer #1 (Recommendations for the authors):**
I highly appreciate the care and detail in the authors' response and thank them for the ePort invested in revising the manuscript. They addressed the core concerns to a high standard, and the manuscript has substantially improved in methodological rigour (through additional controls/sanity checks and complementary mixed-ePects analyses) and in clarity of interpretation (by explicitly acknowledging context-dependence and tempering stronger claims). The present version reads clearly and is much strengthened overall. I only have a few minor points below:Minor suggestions:Abstract:In the abstract KL is introduced as abbreviation, but at first occurence it should be written out as "Kullback-Leibler (KL)" for readers not familiar with it.

We thank the reviewer for catching this error. It has been correct in the version of record.

Methods:I appreciate the additional bayesian LME analysis. I only had a few things that I thought were missing from knowing the parameters: (1) what was the target acceptance rate (default of .95?), (2) which family was used to model the response distribution: (default) "gaussian" or robust "student-t"? Depending on the data a student-t would be preferred, but since the author's checked the fit & the results corroborate the correlation analysis, using the default would also be fine! Just add the information for completeness.

Thank you for bringing this to our attention. We have now noted that default parameters were used in all cases unless otherwise mentioned.

Thank you once again for your time and consideration.

**Reviewer #2 (Recommendations for the authors):**
Thanks to the authors' ePort on revision. I am happy with this new version of manuscript.

Thank you once again for your time and consideration.

**Reviewer #3 (Recommendations for the authors):**
(1) Regarding comments #3 and #6 (first round) on model validation and posterior predictive checks, the authors replied that since their model is not a "generative" one, they can't perform posterior predictive checks. Crucially, in eq. 2, the authors present the p{tilde}^j_k variable denoting the learned probability of event k on trial j. I don't see why this can't be exploited for simulations. In my opinion, one could (and should) generate predictions based on this variable. The simplest implementation would translate the probability into a categorical choice (w/o fitting any free parameter). Based on this, they could assess whether the model and data are comparable.

We thank the reviewer for this clarification. The reviewer suggests using the probability distributions at each trial to predict which event should be chosen on each trial. More specifically, the event(s) with the highest probability on trial j could be used to generate a prediction for the choice of the participant on trial j. We agree that this would indeed be an interesting analysis. However, the response options of each task are limited to two-alternatives. In the cue-target task, four events are modeled (representing all possible cue-target conditions) while the participants’ response options are only “left” and “right”. Similarly, in the letter-color task, 36 events are modeled while the participants’ response options are “match” and “no-match”. In other words, we do not know which event (either four or 36, for the two tasks) the participant would have indicated on each trial. As an approximation to this fine-grained analysis, we investigated the relationship between the information-theoretic variables separately for error and correct trials. Our rationale was that we would have more insight into how the model fits depended on the participants’ actual behavior as compared with the ideal learner model.

(2) I recommend providing a plot of the linear mixed model analysis of the pupil data. Currently, results are only presented in the text and tables, but a figure would be much more useful.

We thank the reviewer for the suggestion to add a plot of the linear mixed model results. We appreciate the value of visualizing model estimates; however, we feel that the current presentation in the text and tables clearly conveys the relevant findings. For this reason, and to avoid further lengthening the manuscript, we prefer to retain the current format.

(3) I would consider only presenting the linear mixed ePects for the pupil data in the main results, and the correlation results in the supplement. It is currently quite long.

We thank the reviewer for this recommendation. We agree that the results section is detailed; however, we consider the correlation analyses to be integral to the interpretation of the pupil data and therefore prefer to keep them in the main text rather than move them to the supplement.

The following is the authors’ response to the original reviews

**eLife Assessment**
This important study seeks to examine the relationship between pupil size and information gain, showing opposite effects dependent upon whether the average uncertainty increases or decreases across trials. Given the broad implications for learning and perception, the findings will be of broad interest to researchers in cognitive neuroscience, decision-making, and computational modelling. Nevertheless, the evidence in support of the particular conclusion is at present incomplete - the conclusions would be strengthened if the authors could both clarify the differences between model-updating and prediction error in their account and clarify the patterns in the data.
**Public Reviews:**

**Reviewer #1 (Public review):**
Summary:This study investigates whether pupil dilation reflects prediction error signals during associative learning, defined formally by Kullback-Leibler (KL) divergence, an information-theoretic measure of information gain. Two independent tasks with different entropy dynamics (decreasing and increasing uncertainty) were analyzed: the cue-target 2AFC task and the lettercolor 2AFC task. Results revealed that pupil responses scaled with KL divergence shortly after feedback onset, but the direction of this relationship depended on whether uncertainty (entropy) increased or decreased across trials. Furthermore, signed prediction errors (interaction between frequency and accuracy) emerged at different time windows across tasks, suggesting taskspecific temporal components of model updating. Overall, the findings highlight that pupil dilation reflects information-theoretic processes in a complex, context-dependent manner.Strengths:This study provides a novel and convincing contribution by linking pupil dilation to informationtheoretic measures, such as KL divergence, supporting Zénon's hypothesis that pupil responses reflect information gained during learning. The robust methodology, including two independent datasets with distinct entropy dynamics, enhances the reliability and generalisability of the findings. By carefully analysing early and late time windows, the authors capture the temporal dynamics of prediction error signals, offering new insights into the timing of model updates. The use of an ideal learner model to quantify prediction errors, surprise, and entropy provides a principled framework for understanding the computational processes underlying pupil responses. Furthermore, the study highlights the critical role of task context - specifically increasing versus decreasing entropy - in shaping the directionality and magnitude of these effects, revealing the adaptability of predictive processing mechanisms.Weaknesses:While this study offers important insights, several limitations remain. The two tasks differ significantly in design (e.g., sensory modality and learning type), complicating direct comparisons and limiting the interpretation of differences in pupil dynamics. Importantly, the apparent context-dependent reversal between pupil constriction and dilation in response to feedback raises concerns about how these opposing effects might confound the observed correlations with KL divergence.

We agree with the reviewer’s concerns and acknowledge that the speculation concerning the directional effect of entropy across trials can not be fully substantiated by the current study. As the reviewer points out, the directional relationship between pupil dilation and information gain must be due to other factors, for instance, the sensory modality, learning type, or the reversal between pupil constriction and dilation across the two tasks. Also, we would like to note that ongoing experiments in our lab already contradict our original speculation. In line with the reviewer’s point, we noted these differences in the section on “Limitations and future research” in the Discussion. To better align the manuscript with the above mentioned points, we have made several changes in the Abstract, Introduction and Discussion summarized below:

We have removed the following text from the Abstract and Introduction: “…, specifically related to increasing or decreasing average uncertainty (entropy) across trials.”

We have edited the following text in the Introduction (changes in italics) (p. 5):

“We analyzed two independent datasets featuring distinct associative learning paradigms, one characterized by increasing entropy and the other by decreasing entropy as the tasks progressed. By examining these different tasks, we aimed to identify commonalities (if any) in the results across varying contexts. Additionally, the contrasting directions of entropy in the two tasks enabled us to disentangle the correlation between stimulus-pair frequency and information gain in the postfeedback pupil response.

We have removed the following text from the Discussion:

“…and information gain in fact seems to be driven by increased uncertainty.”

“We speculate that this difference in the direction of scaling between information gain and the pupil response may depend on whether entropy was increasing or decreasing across trials.”

“…which could explain the opposite direction of the relationship between pupil dilation and information gain”

“… and seems to relate to the direction of the entropy as learning progresses (i.e., either increasing or decreasing average uncertainty).”

We have edited the following texts in the Discussion (changes in italics):

“For the first time, we show that the direction of the relationship between postfeedback pupil dilation and information gain (defined as KL divergence) was context dependent.” (p. 29):

Finally, we have added the following correction to the Discussion (p. 30):

“Although it is tempting to speculate that the direction of the relationship between pupil dilation and information gain may be due to either increasing or decreasing entropy as the task progressed, we must refrain from this conclusion. We note that the two tasks differ substantially in terms of design with other confounding variables and therefore cannot be directly compared to one another. We expand on these limitations in the section below (see Limitations and future research).”

Finally, subjective factors such as participants' confidence and internal belief states were not measured, despite their potential influence on prediction errors and pupil responses.

Thank you for the thoughtful comment. We agree with the reviewer that subjective factors, such as participants' confidence, can be important in understanding prediction errors and pupil responses. As per the reviewer’s point, we have included the following limitation in the Discussion (p. 33):

“Finally, while we acknowledge the potential relevance of subjective factors, such as the participants’ overt confidence reports, in understanding prediction errors and pupil responses, the current study focused on the more objective, model-driven measure of information-theoretic variables. This approach aligns with our use of the ideal learner model, which estimates information-theoretic variables while being agnostic about the observer's subjective experience itself. Future research is needed to explore the relationship between information-gain signals in pupil dilation and the observer’s reported experience of or awareness about confidence in their decisions.”

**Reviewer #2 (Public review):**
Summary:The authors proposed that variability in post-feedback pupillary responses during the associative learning tasks can be explained by information gain, which is measured as KL divergence. They analysed pupil responses in a later time window (2.5s-3s after feedback onset) and correlated them with information-theory-based estimates from an ideal learner model (i.e., information gain-KL divergence, surprise-subjective probability, and entropy-average uncertainty) in two different associative decision-making tasks.Strength:The exploration of task-evoked pupil dynamics beyond the immediate response/feedback period and then associating them with model estimates was interesting and inspiring. This offered a new perspective on the relationship between pupil dilation and information processing.Weakness:However, disentangling these later effects from noise needs caution. Noise in pupillometry can arise from variations in stimuli and task engagement, as well as artefacts from earlier pupil dynamics. The increasing variance in the time series of pupillary responses (e.g., as shown in Figure 2D) highlights this concern.It's also unclear what this complicated association between information gain and pupil dynamics actually means. The complexity of the two different tasks reported made the interpretation more difficult in the present manuscript.

We share the reviewer’s concerns. To make this point come across more clearly, we have added the following text to the Introduction (p. 5):

“The current study was motivated by Zenon’s hypothesis concerning the relationship between pupil dilation and information gain, particularly in light of the varying sources of signal and noise introduced by task context and pupil dynamics. By demonstrating how task context can influence which signals are reflected in pupil dilation, and highlighting the importance of considering their temporal dynamics, we aim to promote a more nuanced and model-driven approach to cognitive research using pupillometry.”

**Reviewer #3 (Public review):**
Summary:This study examines prediction errors, information gain (Kullback-Leibler [KL] divergence), and uncertainty (entropy) from an information-theory perspective using two experimental tasks and pupillometry. The authors aim to test a theoretical proposal by Zénon (2019) that the pupil response reflects information gain (KL divergence). In particular, the study defines the prediction error in terms of KL divergence and speculates that changes in pupil size associated with KL divergence depend on entropy. Moreover, the authors examine the temporal characteristics of pupil correlates of prediction errors, which differed considerably across previous studies that employed different experimental paradigms. In my opinion, the study does not achieve these aims due to several methodological and theoretical issues.Strengths:(1) Use of an established Bayesian model to compute KL divergence and entropy.(2) Pupillometry data preprocessing, including deconvolution.Weaknesses:(1) Definition of the prediction error in terms of KL divergence:I'm concerned about the authors' theoretical assumption that the prediction error is defined in terms of KL divergence. The authors primarily refer to a review article by Zénon (2019): "Eye pupil signals information gain". It is my understanding that Zénon argues that KL divergence quantifies the update of a belief, not the prediction error: "In short, updates of the brain's internal model, quantified formally as the Kullback-Leibler (KL) divergence between prior and posterior beliefs, would be the common denominator to all these instances of pupillary dilation to cognition." (Zénon, 2019).From my perspective, the update differs from the prediction error. Prediction error refers to the difference between outcome and expectation, while update refers to the difference between the prior and the posterior. The prediction error can drive the update, but the update is typically smaller, for example, because the prediction error is weighted by the learning rate to compute the update. My interpretation of Zénon (2019) is that they explicitly argue that KL divergence defines the update in terms of the described difference between prior and posterior, not the prediction error.The authors also cite a few other papers, including Friston (2010), where I also could not find a definition of the prediction error in terms of KL divergence. For example [KL divergence:] "A non-commutative measure of the non-negative difference between two probability distributions." Similarly, Friston (2010) states: Bayesian Surprise - "A measure of salience based on the Kullback-Leibler divergence between the recognition density (which encodes posterior beliefs) and the prior density. It measures the information that can be recognized in the data." Finally, also in O'Reilly (2013), KL divergence is used to define the update of the internal model, not the prediction error.The authors seem to mix up this common definition of the model update in terms of KL divergence and their definition of prediction error along the same lines. For example, on page 4: "KL divergence is a measure of the difference between two probability distributions. In the context of predictive processing, KL divergence can be used to quantify the mismatch between the probability distributions corresponding to the brain's expectations about incoming sensory input and the actual sensory input received, in other words, the prediction error (Friston, 2010; Spratling, 2017)."Similarly (page 23): "In the current study, we investigated whether the pupil's response to decision outcome (i.e., feedback) in the context of associative learning reflects a prediction error as defined by KL divergence."This is problematic because the results might actually have limited implications for the authors' main perspective (i.e., that the pupil encodes prediction errors) and could be better interpreted in terms of model updating. In my opinion, there are two potential ways to deal with this issue:(a) Cite work that unambiguously supports the perspective that it is reasonable to define the prediction error in terms of KL divergence and that this has a link to pupillometry. In this case, it would be necessary to clearly explain the definition of the prediction error in terms of KL divergence and dissociate it from the definition in terms of model updating.(b) If there is no prior work supporting the authors' current perspective on the prediction error, it might be necessary to revise the entire paper substantially and focus on the definition in terms of model updating.

We thank the reviewer for pointy out these inconsistencies in the manuscript and appreciate their suggestions for improvement. We take approach (a) recommended by the reviewer, and provide our reasoning as to why prediction error signals in pupil dilation are expected to correlate with information gain (defined as the KL divergence between posterior and prior belief distributions). This can be found in a new section in the introduction, copied here for convenience (p. 3-4):

“We reasoned that the link between prediction error signals and information gain in pupil dilation is through precision-weighting. Precision refers to the amount of uncertainty (inverse variance) of both the prior belief and sensory input in the prediction error signals [6,64–67]. More precise prediction errors receive more weighting, and therefore, have greater influence on model updating processes. The precisionweighting of prediction error signals may provide a mechanism for distinguishing between known and unknown sources of uncertainty, related to the inherent stochastic nature of a signal versus insufficient information of the part of the observer, respectively [65,67,68]. In Bayesian frameworks, information gain is fundamentally linked to prediction error, modulated by precision [65,66,69–75]. In non-hierarchical Bayesian models, information gain can be derived as a function of prediction errors and the precision of the prior and likelihood distributions, a relationship that can be approximately linear [70]. In hierarchical Bayesian inference, the update in beliefs (posterior mean changes) at each level is proportional to the precision-weighted prediction error; this update encodes the information gained from new observations [65,66,69,71,72]. Neuromodulatory arousal systems are well-situated to act as precision-weighting mechanisms in line with predictive processing frameworks [76,77]. Empirical evidence suggests that neuromodulatory systems broadcast precisionweighted prediction errors to cortical regions [11,59,66,78]. Therefore, the hypothesis that feedback-locked pupil dilation reflects a prediction error signal is similarly in line with Zenon’s main claim that pupil dilation generally reflects information gain, through precision-weighting of the prediction error. We expected a prediction error signal in pupil dilation to be proportional to the information gain.”

We have referenced previous work that has linked prediction error and information gain directly (p. 4): “The KL divergence between posterior and prior belief distributions has been previously considered to be a proxy of (precision-weighted) prediction errors [68,72].”

We have taken the following steps to remedy this error of equating “prediction error” directly with the information gain.

First, we have replaced “KL divergence” with “information gain” whenever possible throughout the manuscript for greater clarity.

Second, we have edited the section in the introduction defining information gain substantially (p. 4):

“Information gain can be operationalized within information theory as the KullbackLeibler (KL) divergence between the posterior and prior belief distributions of a Bayesian observer, representing a formalized quantity that is used to update internal models [29,79,80]. Itti and Baldi (2005)81 termed the KL divergence between posterior and prior belief distributions as “Bayesian surprise” and showed a link to the allocation of attention. The KL divergence between posterior and prior belief distributions has been previously considered to be a proxy of (precision-weighted) prediction errors[68,72]. According to Zénon’s hypothesis, if pupil dilation reflects information gain during the observation of an outcome event, such as feedback on decision accuracy, then pupil size will be expected to increase in proportion to how much novel sensory evidence is used to update current beliefs [29,63]. ”

Finally, we have made several minor textual edits to the Abstract and main text wherever possible to further clarify the proposed relationship between prediction errors and information gain.

(2) Operationalization of prediction errors based on frequency, accuracy, and their interaction:The authors also rely on a more model-agnostic definition of the prediction error in terms of stimulus frequency ("unsigned prediction error"), accuracy, and their interaction ("signed prediction error"). While I see the point here, I would argue that this approach offers a simple approximation to the prediction error, but it is possible that factors like difficulty and effort can influence the pupil signal at the same time, which the current approach does not take into account. I recommend computing prediction errors (defined in terms of the difference between outcome and expectation) based on a simple reinforcement-learning model and analyzing the data using a pupillometry regression model in which nuisance regressors are controlled, and results are corrected for multiple comparisons.

We agree with the reviewer’s suggestion that alternatively modeling the data in a reinforcement learning paradigm would be fruitful. We adopted the ideal learner model as we were primarily focused on Information Theory, stemming from our aim to test Zenon’s hypothesis that information gain drives pupil dilation. However, we agree with the reviewer that it is worthwhile to pursue different modeling approaches in future work. We have now included a complementary linear mixed model analysis in which we controlled for the effects of the information-theoretic variables on one another, while also including the nuisance regressors of pre-feedback baseline pupil dilation and reaction times (explained in more detail below in our response to your point #4). Results including correction for multiple comparisons was reported for all pupil time course data as detailed in Methods section 2.5.

(3) The link between model-based (KL divergence) and model-agnostic (frequency- and accuracy-based) prediction errors:I was expecting a validation analysis showing that KL divergence and model-agnostic prediction errors are correlated (in the behavioral data). This would be useful to validate the theoretical assumptions empirically.

The model limitations and the operalization of prediction error in terms of post-feedback processing do not seem to allow for a comparison of information gain and model-agnostic prediction errors in the behavioral data for the following reasons. First, the simple ideal learner model used here is not a generative model, and therefore, cannot replicate or simulate the participants responses (see also our response to your point #6 “model validation” below). Second, the behavioral dependent variables obtained are accuracy and reaction times, which both occur before feedback presentation. While accuracy and reaction times can serve as a marker of the participant’s (statistical) confidence/uncertainty following the decision interval, these behavioral measures cannot provide access to post-feedback information processing. The pupil dilation is of interest to us because the peripheral arousal system is able to provide a marker of post-feedback processing. Through the analysis presented in Figure 3, we indeed aimed to make the comparison of the model-based information gain to the model-agnostic prediction errors via the proxy variable of post-feedback pupil dilation instead of behavioral variables. To bridge the gap between the “behaviorally agnostic” model parameters and the actual performance of the participants, we examined the relationship between the model-based information gain and the post-feedback pupil dilation separately for error and correct trials as shown in Figure 3D-F & Figure 3J-L. We hope this addresses the reviewers concern and apologize in case we did not understand the reviewers suggestion here.

(4) Model-based analyses of pupil data:I'm concerned about the authors' model-based analyses of the pupil data. The current approach is to simply compute a correlation for each model term separately (i.e., KL divergence, surprise, entropy). While the authors do show low correlations between these terms, single correlational analyses do not allow them to control for additional variables like outcome valence, prediction error (defined in terms of the difference between outcome and expectation), and additional nuisance variables like reaction time, as well as x and y coordinates of gaze.Moreover, including entropy and KL divergence in the same regression model could, at least within each task, provide some insights into whether the pupil response to KL divergence depends on entropy. This could be achieved by including an interaction term between KL divergence and entropy in the model.

In line with the reviewer’s suggestions, we have included a complementary linear mixed model analysis in which we controlled for the effects of the information-theoretic variables on one another, while also including the nuisance regressors of pre-feedback baseline pupil dilation and reaction times. We compared the performance of two models on the post-feedback pupil dilation in each time window of interest: Modle 1 had no interaction between information gain and entropy and Model 2 included an interaction term as suggested. We did not include the x- and y- coordinates of gaze in the mixed linear model analysis, as there are multiple values of these coordinates per trial. Furthermore, regressing out the x and y- coordinates of gaze can potentially remove signal of interest in the pupil dilation data in addition to the gaze-related confounds and we did not measure absolute pupil size (Mathôt, Melmi & Castet, 2015; Hayes & Petrov, 2015). We present more sanity checks on the pre-processing pipeline as recommended by Reviewer 1.

This new analysis resulted in several additions to the Methods (see Section 2.5) and Results. In sum, we found that including an interaction term for information gain and entropy did not lead to better model fits, but sometimes lead to significantly worse fits. Overall, the results of the linear mixed model corroborated the “simple” correlation analysis across the pupil time course while accounting for the relationship to the pre-feedback baseline pupil and preceeding reaction time differences. There was only one difference to note between the correlation and linear mixed modeling analyses: for the error trials in the cue-target 2AFC task, including entropy in the model accounted for the variance previously explained by surprise.

(5) Major differences between experimental tasks:More generally, I'm not convinced that the authors' conclusion that the pupil response to KL divergence depends on entropy is sufficiently supported by the current design. The two tasks differ on different levels (stimuli, contingencies, when learning takes place), not just in terms of entropy. In my opinion, it would be necessary to rely on a common task with two conditions that differ primarily in terms of entropy while controlling for other potentially confounding factors. I'm afraid that seemingly minor task details can dramatically change pupil responses. The positive/negative difference in the correlation with KL divergence that the authors interpret to be driven by entropy may depend on another potentially confounding factor currently not controlled.

We agree with the reviewer’s concerns and acknowledge that the speculation concerning the directional effect of entropy across trials can not be fully substantiated by the currect study. We note that Review #1 had a similar concern. Our response to Reviewer #1 addresses this concern of Reviewer #3 as well. To better align the manuscript with the above mentioned points, we have made several changes that are detailed in our response to Reviewer #1’s public review (above).

(6) Model validation:My impression is that the ideal learner model should work well in this case. However, the authors don't directly compare model behavior to participant behavior ("posterior predictive checks") to validate the model. Therefore, it is currently unclear if the model-derived terms like KL divergence and entropy provide reasonable estimates for the participant data.

Based on our understanding, posterior predictive checks are used to assess the goodness of fit between generated (or simulated) data and observed data. Given that the “simple” ideal learner model employed in the current study is not a generative model, a posterior predictive check would not apply here (Gelman, Carlin, Stern, Dunson, Vehtari, & Rubin 2013). The ideal learner model is unable to simulate or replicate the participants’ responses and behaviors such as accuracy and reaction times; it simply computes the probability of seeing each stimulus type at each trial based on the prior distribution and the exact trial order of the stimuli presented to each participant. The model’s probabilities are computed directly from a Dirichlet distribution of values that represent the number of occurences of each stimulus-pair type for each task. The information-theoretic variables are then directly computed from these probabilities using standard formulas. The exact formulas used in the ideal learner model can be found in section 2.4.

We have now included a complementary linear mixed model analysis which also provides insight into the amount of explained variance of these information-theoretic predictors on the post-feedback pupil response, while also including the pre-feedback baseline pupil and reaction time differences (see section 3.3, Tables 3 & 4). The R^2^ values ranged from 0.16 – 0.50 across all conditions tested.

(7) Discussion:The authors interpret the directional effect of the pupil response w.r.t. KL divergence in terms of differences in entropy. However, I did not find a normative/computational explanation supporting this interpretation. Why should the pupil (or the central arousal system) respond differently to KL divergence depending on differences in entropy?The current suggestion (page 24) that might go in this direction is that pupil responses are driven by uncertainty (entropy) rather than learning (quoting O'Reilly et al. (2013)). However, this might be inconsistent with the authors' overarching perspective based on Zénon (2019) stating that pupil responses reflect updating, which seems to imply learning, in my opinion. To go beyond the suggestion that the relationship between KL divergence and pupil size "needs more context" than previously assumed, I would recommend a deeper discussion of the computational underpinnings of the result.

Since we have removed the original speculative conclusion from the manuscript, we will refrain from discussing the computational underpinnings of a potential mechanism. To note as mentioned above, we have preliminary data from our own lab that contradicts our original hypothesis about the relationship between entropy and information gain on the post-feedback pupil response.

**Recommendations for the authors:**

**Reviewer #1 (Recommendations for the authors):**
Apart from the points raised in the public review above, I'd like to use the opportunity here to provide a more detailed review of potential issues, questions, and queries I have:(1) Constriction vs. Dilation Effects:The study observes a context-dependent relationship between KL divergence and pupil responses, where pupil dilation and constriction appear to exhibit opposing effects. However, this phenomenon raises a critical concern: Could the initial pupil constriction to visual stimuli (e.g., in the cue-target task) confound correlations with KL divergence? This potential confound warrants further clarification or control analyses to ensure that the observed effects genuinely reflect prediction error signals and are not merely a result of low-level stimulus-driven responses.

We agree with the reviewers concern and have added the following information to the limitations section in the Discussion (changes in italics below; p. 32-33).

“First, the two associative learning paradigms differed in many ways and were not directly comparable. For instance, the shape of the mean pupil response function differed across the two tasks in accordance with a visual or auditory feedback stimulus (compare Supplementary Figure 3A with Supplementary Figure 3D), and it is unclear whether these overall response differences contributed to any differences obtained between task conditions within each task. We are unable to rule out whether so-called “low level” effects such as the initial constriction to visual stimuli in the cue-target 2AFC task as compared with the dilation in response auditory stimuli in letter-color 2AFC task could confound correlations with information gain. Future work should strive to disentangle how the specific aspects of the associative learning paradigms relate to prediction errors in pupil dilation by systematically manipulating design elements within each task.”

Here, I also was curious about Supplementary Figure 1, showing 'no difference' between the two tones (indicating 'error' or 'correct'). Was this the case for FDR-corrected or uncorrected cluster statistics? Especially since the main results also showed sig. differences only for uncorrected cluster statistics (Figure 2), but were n.s. for FDR corrected. I.e. can we be sure to rule out a confound of the tones here after all?

As per the reviewer’s suggestion, we verified that there were also no significant clusters after feedback onset before applying the correction for multiple comparisons. We have added this information to Supplemenatary section 1.2 as follows:

“Results showed that the auditory tone dilated pupils on average (Supplementary Figure 1C). Crucially, however, the two tones did not differ from one another in either of the time windows of interest (Supplementary Figure 1D); no significant time points after feedback onset were obtained either before or after correcting for multiple comparisons using cluster-based permutation methods; see Section 2.5.”

Supplementary Figure 1 is showing effects cluster-corrected for multiple comparisons using cluster-based permutation tests from the MNE software package in Python (see Methods section 2.5). We have clarified that the cluster-correction was based on permutation testing in the figure legend.

(2) Participant-Specific Priors:The ideal learner models do not account for individualised priors, assuming homogeneous learning behaviour across participants. Could incorporating participant-specific priors better reflect variability in how individuals update their beliefs during associative learning?

We have clarified in the Methods (see section 2.4) that the ideal learner models did account for participant-specific stimuli including participant-specific priors in the letter-color 2AFC task. We have added the following texts:

“We also note that while the ideal learner model for the cue-target 2AFC task used a uniform (flat) prior distribution for all participants, the model parameters were based on the participant-specific cue-target counterbalancing conditions and randomized trial order.” (p. 13)

“The prior distributions used for the letter-color 2AFC task were estimated from the randomized letter-color pairs and randomized trial order presentation in the preceding odd-ball task; this resulted in participant-specific prior distributions for the ideal learner model of the letter-color 2AFC task. The model parameters were likewise estimated from the (participant-specific) randomized trial order presented in the letter-color 2AFC task.” (p. 13)

(3) Trial-by-Trial Variability:The analysis does not account for random effects or inter-trial variability using mixed-effects models. Including such models could provide a more robust statistical framework and ensure the observed relationships are not influenced by unaccounted participant- or trial-specific factors.

We have included a complementary linear mixed model analysis in which “subject” was modeled as a random effect on the post-feedback pupil response in each time window of interest and for each task. Across all trials, the results of the linear mixed model corroborated the “simple” correlation analysis across the pupil time course while accounting for the relationship to the prefeedback baseline pupil and preceeding reaction time differences (see section 3.3, Tables 3 & 4).

(4) Preprocessing/Analysis choices:Before anything else, I'd like to highlight the authors' effort in providing public code (and data) in a very readable and detailed format!

We appreciate the compliment - thank you for taking the time to look at the data and code provided.

I found the idea of regressing the effect of Blinks/Saccades on the pupil trace intriguing. However, I miss a complete picture here to understand how well this actually worked, especially since it seems to be performed on already interpolated data. My main points here are:(4.1) Why is the deconvolution performed on already interpolated data and not on 'raw' data where there are actually peaks of information to fit?

To our understanding, at least one critical reason for interpolating the data before proceeding with the deconvolution analysis is that the raw data contain many missing values (i.e., NaNs) due to the presence of blinks. Interpolating over the missing data first ensures that there are valid numerical elements in the linear algebra equations. We refer the reviewer to the methods detailed in Knapen et al. (2016) for more details on this pre-processing method.

(4.2) What is the model fit (e.g. R-squared)? If this was a poor fit for the regressors in the first place, can we trust the residuals (i.e. clean pupil trace)? Is it possible to plot the same Pupil trace of Figure 1D with (a) the 'raw' pupil time-series, (b) after interpolation only (both of course also mean-centered for comparison), on top of the residuals after deconvolution (already presented), so we can be sure that this is not driving the effects in a 'bad' way? I'd just like to make sure that this approach did not lead to artefacts in the residuals rather than removing them.

We thank the reviewer for this suggestion. In the Supplementary Materials, we have included a new figure (Supplementary Figure 2, copied below for convience), which illustrates the same conditions as in Figure 1D and Figure 2D, with (1) the raw data, and (2) the interpolated data before the nuisance regression. Both the raw data and interpolated data have been band-pass filtered as was done in the original pre-processing pipeline and converted to percent signal change. These figures can be compared directly to Figure 1D and Figure 2D, for the two tasks, respectively.

Of note is that the raw data seem to be dominated by responses to blinks (and/or saccades). Crucially, the pattern of results remains overall unchaged between the interpolated-only and fully pre-processed version of the data for both tasks.

In the Supplementary Materials (see Supplementary section 2), we have added the descriptives of the model fits from the deconvolution method. Model fits (R^2^) for the nuisance regression were generally low: cue-target 2AFC task, M = 0.03, SD = 0.02, range = [0.00, 0.07]; letter-color visual 2AFC, M = 0.08, SD = 0.04, range = [0.02, 0.16].

Furthermore, a Pearson correlation analysis between the interpolated and fully pre-processed data within the time windows of interest for both task indicated high correspondence:

Cue-target 2AFC task

Early time window: M = 0.99, SD = 0.01, range = [0.955, 1.000]

Late time window: M = 0.99, SD = 0.01, range = [0.971, 1.000]

Letter-color visual 2AFC

Early time window: M = 0.95, SD = 0.04, range = [0.803, 0.998]

Late time window: M = 0.97, SD = 0.02, range = [0.908, 0.999]

In hindsight, including the deconvolution (nuisance regression) method may not have changed the pattern of results much. However, the decision to include this deconvolution method was not data-driven; instead, it was based on the literature establishing the importance of removing variance (up to 5 s) of these blinks and saccades from cognitive effects of interest in pupil dilation (Knapen et al., 2016).

(4.3) Since this should also lead to predicted time series for the nuisance-regressors, can we see a similar effect (of what is reported for the pupil dilation) based on the blink/saccade traces of (a) their predicted time series based on the deconvolution, which could indicate a problem with the interpretation of the pupil dilation effects, and (b) the 'raw' blink/saccade events from the eye-tracker? I understand that this is a very exhaustive analysis so I would actually just be interested here in an averaged time-course / blink&saccade frequency of the same time-window in Figure 1D to complement the PD analysis as a sanity check.

Also included in the Supplementary Figure 2 is the data averaged as in Figure 1D and Figure 2D for the raw data and nuisance-predictor time courses (please refer to the bottom row of the sub-plots). No pattern was observed in either the raw data or the nuisance predictors as was shown in the residual time courses.

(4.4) How many samples were removed from the time series due to blinks/saccades in the first place? 150ms for both events in both directions is quite a long bit of time so I wonder how much 'original' information of the pupil was actually left in the time windows of interest that were used for subsequent interpretations.

We thank the reviewer for bringing this issue to our attention. The size of the interpolation window was based on previous literature, indicating a range of 100-200 ms as acceptable (Urai et al., 2017; Knapen et al., 2016; Winn et al., 2018). The ratio of interpolated-to-original data (across the entire trial) varied greatly between participants and between trials: cue-target 2AFC task, M = 0.262, SD = 0.242, range = [0,1]; letter-color 2AFC task, M = 0.194, SD = 0.199, range = [0,1].

We have now included a conservative analysis in which only trials with more than half (threshold = 60%) of original data are included in the analyses. Crucially, we still observe the same pattern of effects as when all data are considered across both tasks (compare the second to last row in the Supplementary Figure 2 to Figure 1D and Figure 2D).

(4.5) Was the baseline correction performed on the percentage change unit?

Yes, the baseline correction was performed on the pupil timeseries after converting to percentsignal change. We have added that information to the Methods (section 2.3).

(4.6) What metric was used to define events in the derivative as 'peaks'? I assume some sort of threshold? How was this chosen?

The threshold was chosen in a data-driven manner and was kept consistent across both tasks. The following details have been added to the Methods:

“The size of the interpolation window preceding nuisance events was based on previous literature [13,39,99]. After interpolation based on data-markers and/or missing values, remaining blinks and saccades were estimated by testing the first derivative of the pupil dilation time series against a threshold rate of change. The threshold for identifying peaks in the temporal derivative is data-driven, partially based on past work[10,14,33]. The output of each participant’s pre-processing pipeline was checked visually. Once an appropriate threshold was established at the group level, it remained the same for all participants (minimum peak height of 10 units).” (p. 8 & 11).

(5) Multicollinearity Between Variables:Lastly, the authors state on page 13: "Furthermore, it is expected that these explanatory variables will be correlated with one another. For this reason, we did not adopt a multiple regression approach to test the relationship between the information-theoretic variables and pupil response in a single model". However, the very purpose of multiple regression is to account for and disentangle the contributions of correlated predictors, no? I might have missed something here.

We apologize for the ambiguity of our explanation in the Methods section. We originally sought to assess the overall relationship between the post-feedback response and information gain (primarily), but also surprise and entropy. Our reasoning was that these variables are often investigated in isolation across different experiments (i.e., only investigating Shannon surprise), and we would like to know what the pattern of results would look like when comparing a single information-theoretic variable to the pupil response (one-by-one). We assumed that including additional explanatory variables (that we expected to show some degree of collinearity with each other) in a regression model would affect variance attributed to them as compared with the one-on-one relationships observed with the pupil response (Morrissey & Ruxton 2018). We also acknowledge the value of a multiple regression approach on our data. Based on the suggestions by the reviewers we have included a complementary linear mixed model analysis in which we controlled for the effects of the information-theoretic variables on one another, while also including the nuisance regressors of pre-feedback baseline pupil dilation and reaction times.

This new analysis resulted in several additions to the Methods (see Section 2.5) and Results (see Tables 3 and 4). Overall, the results of the linear mixed model corroborated the “simple” correlation analysis across the pupil time course while accounting for the relationship to the prefeedback baseline pupil and preceeding reaction time differences. There was only one difference to note between the correlation and linear mixed modeling analyses: for the error trials in the cue-target 2AFC task, including entropy in the model accounted for the variance previously explained by surprise.

**Reviewer #2 (Recommendations for the authors):**
(1) Given the inherent temporal dependencies in pupil dynamics, characterising later pupil responses as independent of earlier ones in a three-way repeated measures ANOVA may not be appropriate. A more suitable approach might involve incorporating the earlier pupil response as a covariate in the model.

We thank the reviewer for bringing this issue to our attention. From our understanding, a repeated-measures ANOVA with factor “time window” would be appropriate in the current context for the following reasons. First, autocorrelation (closely tied to sphericity) is generally not considered a problem when only two timepoints are compared from time series data (Field, 2013; Tabachnick & Fidell, 2019). Second, the repeated-measures component of the ANOVA takes the correlated variance between time points into account in the statistical inference. Finally, as a complementary analysis, we present the results testing the interaction between the frequency and accuracy conditions across the full time courses (see Figures 1D and 2D); in these pupil time courses, any difference between the early and late time windows can be judged by the reader visually and qualitatively.

(2) Please clarify the correlations between KL divergence, surprise, entropy, and pupil response time series. Specifically, state whether these correlations account for the interrelationships between these information-theoretic measures. Given their strong correlations, partialing out these effects is crucial for accurate interpretation.

As mentioned above, based on the suggestions by the reviewers we have included a complementary linear mixed model analysis in which we controlled for the effects of the information-theoretic variables on one another, while also including the nuisance regressors of pre-feedback baseline pupil dilation and reaction times.

This new analysis resulted in several additions to the Methods (see Section 2.5) and Results (see Tables 3 and 4). Overall, the results of the linear mixed model corroborated the “simple” correlation analysis across the pupil time course while accounting for the relationship to the prefeedback baseline pupil and preceeding reaction time differences. There was only one difference to note between the correlation and linear mixed modeling analyses: for the error trials in the cue-target 2AFC task, including entropy in the model accounted for the variance previously explained by surprise.

(3) The effects observed in the late time windows appear weak (e.g., Figure 2E vs. 2F, and the generally low correlation coefficients in Figure 3). Please elaborate on the reliability and potential implications of these findings.

We have now included a complementary linear mixed model analysis which also provides insight into the amount of explained variance of these information-theoretic predictors on the post-feedback pupil response, while also including the pre-feedback baseline pupil and reaction time differences (see section 3.3, Tables 3 & 4). The R^2^ values ranged from 0.16 – 0.50 across all conditions tested. Including the pre-feedback baseline pupil dilation as a predictor in the linear mixed model analysis consistently led to more explained variance in the post-feedback pupil response, as expected.

(4) In Figure 3 (C-J), please clarify how the trial-by-trial correlations were computed (averaged across trials or subjects). Also, specify how the standard error of the mean (SEM) was calculated (using the number of participants or trials).

The trial-by-trial correlations between the pupil signal and model parameters were computed for each participant, then the coefficients were averaged across participants for statistical inference. We have added several clarifications in the text (see section 2.5 and legends of Figure 3 and Supplementary Figure 4).

We have added “the standard error of the mean across participants” to all figure labels.

(5) For all time axes (e.g., Figure 2D), please label the ticks at 0, 0.5, 1, 1.5, 2, 2.5, and 3 seconds. Clearly indicate the duration of the feedback on the time axes. This is particularly important for interpreting the pupil dilation responses evoked by auditory feedback.

We have labeled the x-ticks every 0.5 seconds in all figures and indicated the duration of the auditory feedback in the letter-color decision task and as well as the stimuli presented in the control tasks in the Supplementary Materials.

**Reviewer #3 (Recommendations for the authors):**
(1) Introduction page 3: "In information theory, information gain quantifies the reduction of uncertainty about a random variable given the knowledge of another variable. In other words, information gain measures how much knowing about one variable improves the prediction or understanding of another variable."(2) In my opinion, the description of information gain can be clarified. Currently, it is not very concrete and quite abstract. I would recommend explaining it in the context of belief updating.

We have removed these unclear statements in the Introduction. We now clearly state the following:

“Information gain can be operationalized within information theory as the KullbackLeibler (KL) divergence between the posterior and prior belief distributions of a Bayesian observer, representing a formalized quantity that is used to update internal models [29,79,80].” (p. 4)

(3) Page 4: The inconsistencies across studies are described in extreme detail. I recommend shortening this part and summarizing the inconsistencies instead of listing all of the findings separately.

As per the reviewer’s recommendation, we have shortened this part of the introduction to summarize the inconsistencies in a more concise manner as follows:

“Previous studies have shown different temporal response dynamics of prediction error signals in pupil dilation following feedback on decision outcome: While some studies suggest that the prediction error signals arise around the peak (~1 s) of the canonical impulse response function of the pupil [11,30,41,61,62,90], other studies have shown evidence that prediction error signals (also) arise considerably later with respect to feedback on choice outcome [10,25,32,41,62]. A relatively slower prediction error signal following feedback presentation may suggest deeper cognitive processing, increased cognitive load from sustained attention or ongoing uncertainty, or that the brain is integrating multiple sources of information before updating its internal model. Taken together, the literature on prediction error signals in pupil dilation following feedback on decision outcome does not converge to produce a consistent temporal signature.” (p. 5)

We would like to note some additional minor corrections to the preprint:

We have clarified the direction of the effect in Supplementary Figure 3 with the following:

“Participants who showed a larger mean difference between the 80% as compared with the 20% frequency conditions in accuracy also showed smaller differences (a larger mean difference in magnitude in the negative direction) in pupil responses between frequency conditions (see Supplementary Figure 4).”

The y-axis labels in Supplementary Figure 3 were incorrect and have been corrected as the following: “Pupil responses (80-20%)”.

We corrected typos, formatting and grammatical mistakes when discovered during the revision process. Some minor changes were made to improve clarity. Of course, we include a version of the manuscript with Tracked Changes as instructed for consideration.